



# High-resolution regional emission inventory contributes to the evaluation of policy effectiveness: A case study in Jiangsu province, China

Chen Gu[1], Lei Zhang[1,2], Zidie Xu[1], Sijia Xia[3], Yutong Wang[1], Li Li[3], Zeren Wang[1],
Qiuyue Zhao[3], Hanying Wang[1], Yu Zhao[1,2*]
[1] State Key Laboratory of Pollution Control and Resource Reuse and School of the
Environment, Nanjing University, 163 Xianlin Rd., Nanjing, Jiangsu 210023, China
[2] Collaborative Innovation Center of Atmospheric Environment and Equipment
Technology, CICAEET, Nanjing, Jiangsu 210044, China
[3] Jiangsu Key Laboratory of Environmental Engineering, Jiangsu Provincial Academy
of Environmental Sciences, Nanjing, Jiangsu 210036, China
*Corresponding author: Yu Zhao
Phone: 86-25-89680650; email: yuzhao@nju.edu.cn



## Abstract

China has been conducting a series of actions on air quality improvement for the past
decades, and air pollutant emissions have been changing swiftly across the country.
Province is an important administrative unit for air quality management, thus reliable
provincial-level emission inventory for multiple years is essential for detecting the
varying sources of pollution and evaluating the effectiveness of emission controls. In
this study, we selected Jiangsu, one of the most developed provinces in China, and
developed the high-resolution emission inventory of nine species for 2015-2019, with
improved methodologies for different emission sectors, best available facility-level
information on individual sources, and real-world emission measurements. Resulting
from implementation of strict emission control measures, the anthropogenic emissions
were estimated to have declined 53%, 20%, 7%, 2%, 10%, 21%, 16%, 6% and 18%
for $SO_2$, $NO_X$, CO, NMVOCs, $NH_3$, $PM_{10}$, $PM_{2.5}$, BC, and OC from 2015 to 2019,
respectively. Larger abatement of $SO_2$, $NO_X$ and $PM_{2.5}$ emissions were detected for
the more developed southern Jiangsu. Since 2016, the ratio of biogenic volatile
organic compounds (BVOCs) to anthropogenic volatile organic compounds (AVOCs)
exceeded 50% in July, indicating the importance of biogenic sources on summer $O_3$
formation. Our estimates in annual emissions of $NO_X$, NMVOCs, and $NH_3$ were
generally smaller than the national emission inventory MEIC, but larger for primary
particles. The discrepancies between studies resulted mainly from different methods
of emission estimation (e.g., the procedure-based approach for AVOCs emissions
from key industries used in this work) and inconsistent information of emission
source operation (e.g., the penetrations and removal efficiencies of air pollution
control devices). Regarding the different periods, more reduction of $SO_2$ emissions
was found between 2015 and 2017, but $NO_X$, AVOCs and $PM_{2.5}$ between 2017 and
2019. Among the selected 13 major measures, the ultra-low emission retrofit on
power sector was the most important contributor to the reduced $SO_2$ and $NO_X$
emissions (accounting for 38% and 43% of the emission abatement, respectively) for
2015-2017, but its effect became very limited afterwards as the retrofit had been



commonly completed by 2017. Instead, extensive management of coal-fired boilers
and upgradation and renovation of non-electrical industry were the most important
measures for 2017-2019, accounted collectively for 61%, 49% and 57% reduction of
$SO_2$, $NO_X$ and $PM_{2.5}$, respectively. Controls on key industrial sectors maintained the
most effective for AVOCs reduction for the two periods, while measures on other
sources (transportation and solvent replacement) became increasingly important for
more recent years. Our provincial emission inventory was demonstrated to be
supportive for high-resolution air quality modeling for multiple years. Through
scenario setting and modeling, worsened meteorological conditions were found from
2015 to 2019 for $PM_{2.5}$ and $O_3$ pollution alleviation. However, the efforts on emission
controls were identified to largely overcome the negative influence of meteorological
variation. The changed anthropogenic emissions were estimated to contribute 4.3 and
5.5 $\mu g \cdot m^{-3}$ of $PM_{2.5}$ concentration reduction for 2015-2017 and 2017-2019,
respectively. While elevated $O_3$ by 4.9 $\mu g \cdot m^{-3}$ for 2015-2017, the changing emissions
led to 3.1 $\mu g \cdot m^{-3}$ of reduction for 2017-2019, partly (not fully though) offsetting the
meteorology-driven growth. The analysis justified the validity of local emission
control efforts on air quality improvement, and provided scientific basis to formulate
air pollution prevention and control policies for other developed regions in China and
worldwide.

## 1. Introduction

Severe air pollution is of great concern for fast industrialized countries like China,
especially in economically developed regions where an overlap of serious pollution
levels and dense populations has resulted in high exposure and adverse health
outcomes (Klimont et al., 2013; Hoesly et al., 2018). Emission inventory, which
contains complete information on magnitude, spatial pattern, and temporal change of
air pollutant emissions by sector, is essential for identifying the sources of air
pollution and effectiveness of emission controls on air quality through numerical
modeling (Zhao et al., 2013). Improving the understanding of emission behaviors and



reducing the uncertainty of emission estimates have always been the main focus of
emission inventory studies, given the big variety of source categories, fast changing
mix of manufacturing and emission control technologies, and insufficient
measurements of real-world emissions. At the global and continental scales, emission
inventories have been developed by combining available information of large point
sources and improved surrogate statistics for area sources, e.g., Emissions Database
for Global Atmospheric Research (EDGAR, https://edgar.jrc.ec.europa.eu/, Crippa et
al., 2020) and Regional Emission Inventory in Asia (REAS,
https://www.nies.go.jp/REAS/, Kurokawa et al., 2020). As the largest developing
country in the world, China has been proven to contribute significantly to global
emissions (Klimont et al., 2013; Huang et al., 2014; Wiedinmyer et al., 2014;
Miyazaki et al., 2017).
Along with the gradually improved methodology and increasingly availability of
emission source and field measurement data, the applicability and reliability of recent
Chinese emission inventories (e.g., the Multi-resolution Emission Inventory for China,
MEIC, Zheng et al., 2018) have been significantly improved compared to the earlier
large-scale studies for Asia or the world. When the research focus switches to smaller
provincial and city scales, the uncertainty of national emission inventory may increase
attributed mainly to the insufficient information on detailed emission sources,
particularly for medium/small size stationary and area sources. Certain "proxies"
including population and economic densities were commonly applied to downscale
the emissions from coarser to finer horizontal resolution, based on the assumption that
those proxies were strongly associated with emission intensity. Such "coupling effect",
however, has been demonstrated to be largely weakened, leading to great uncertainty
in emission estimation and consequently enhanced bias in air quality modeling (Zhou
et al., 2017; Zheng et al., 2017). For the urgent demand for preventing regional air
pollution and relevant health damage, therefore, development of high-resolution
emission inventories has been getting increasingly essential, especially in regions with
developed industry, large population and complex emission sources (Zheng et al.,
2009; Shen et al., 2017; Zhao et al., 2018). With increased proportion of point sources





and more complete facility-based information, the improved emission inventory could
largely reduce the arbitrary use of proxy-based downscaling technique and thereby the
uncertainty of the emission estimates (Zhao et al., 2015; Zheng et al., 2021).
For the past decade, China has been conducting a series of actions to tackle the
serious air pollution problem. With the mitigation of severe fine particulate matter
($PM_{2.5}$) pollution set as a priority from 2013 to 2017, the National Action Plan on Air
Pollution Control and Prevention (NAPAPCP, State Council of the People's Republic
of China (SCC), 2013) pushed stringent end-of-pipe emission controls (e.g., the
"ultra-low" emission control for power sector) and retirement of small and
energy-inefficient factories (Zhang et al., 2019a; 2019b; Zheng et al., 2018). On top of
that, China announced the "Three-Year Action Plan to Fight Air Pollution"
(TYAPFAP) to further reduce $PM_{2.5}$ and ozone ($O_3$) levels for 2018-2020 (SCC, 2018).
Substantially enhanced measures have been required for reducing industrial (e.g.,
application of "ultra-low" emission control for selected non-electrical industries) and
residential emissions (e.g., promotion of advanced stoves and clean coal during
heating seasons). Those measures have significantly changed the air pollutant
emissions and thereby air quality over the country. Studies have been conducted to
assess the contribution of the nation actions to the improvement of air quality, based
usually on the national emission inventory. For example, Zhang et al. (2019a)
estimated a nationwide 30-40% reduction in $PM_{2.5}$ concentration attributed to
NAPAPCP from 2013 to 2017.
Province is an important administrative unit for air quality management. Given the
heterogeneous economical and energy structures as well as atmospheric conditions,
there are usually big diversities in the strategies and actions of reducing regional air
pollution adopted by the local governments, leading to various progresses of emission
and air quality changes (Liu et al., 2022; Wang et al., 2021a). Limited by incomplete
or inconsecutive information on emission sources and lack of on-time emission
measurements, however, there were relatively few studies on provincial-level
emission inventories for multiple years. Studies based on the national emission
inventories would be less supportive for policy makers to formulate the emission



control measures and to evaluate their effectiveness on emission reduction and air
quality improvement (An et al., 2021; Huang et al., 2021). Contrary to NAPAPCP that
has been increasingly noticed, moreover, few analyses have been conducted for
TYAPFAP after 2017 due partly to lack of most recent emission data, preventing
comparison and comprehensive understanding of the effectiveness of emission
controls for the two phases. Jiangsu Province, located on the northeast coast of the
Yangtze River Delta region (YRD), is one of China's most industrial developed and
heavy-polluted regions. It comprised 10.1% of the gross domestic product (GDP) in
mainland China (ranking the second place in the country), and 6.4%, 11.3% and 11.4%
of national cement, pig iron and crude steel production in 2020, respectively (National
Bureau of Statistics of China, 2021). MEIC indicated the emissions per unit area of
anthropogenic sulfur dioxide ($SO_2$), nitrogen oxides ($NO_X$), non-methane volatile
organic compounds (NMVOCs), $PM_{2.5}$, and ammonia ($NH_3$) in Jiangsu were 2.8, 6.5,
7.0, 4.5 and 4.8 times of the national average in 2017, respectively. Resulting from the
implementation of air pollution prevention measures, $PM_{2.5}$ pollution in Jiangsu has
been significantly alleviated since 2013, while the great changes in emissions due to
varying energy use and industry and transportation development have made it become
the province with the highest $O_3$ concentration and the fastest growth rate of $O_3$ in
YRD for recent years (Zheng et al., 2016; Wang et al., 2017; Zhang et al., 2017a;
Zhou et al., 2017).
In this study, therefore, we took Jiangsu as an example to demonstrate the
development of high-resolution emission inventory and its application on evaluating
the effectiveness of emission control actions. We integrated the methodological
improvements on regional emission inventory by our previous studies (Zhou et al.,
2017; Zhao et al., 2017; 2020; Wu et al., 2022; Zhang et al., 2019b; Zhang et al., 2020;
2021b), and compiled and incorporated best available facility-level information and
real-world emission measurements (see details in the methodology and data section).
A provincial-level emission inventory for 2015-2019 was then thoroughly developed
for nine gaseous and particulate species ($SO_2$, $NO_X$, NMVOCs, carbon dioxide (CO),
inhalable particulate matter ($PM_{10}$), $PM_{2.5}$, $NH_3$, black carbon (BC), and organic





carbon (OC)). The difference between our emission inventory and others, as well as
its main causes, was carefully explored. Using a measure-specific integrated
evaluation approach, we further identified the drivers of emission changes of $SO_2$,
$NO_X$, $PM_{2.5}$ and anthropogenic volatile organic compounds (AVOCs), with an
emphasis on the impacts of 13 major control measures summarized from NAPAPCP
and TYAPFAP. Finally, air quality modeling was applied to assess the reliability of
our emission inventory and to quantify the contribution of emission controls to the
changing $PM_{2.5}$ and $O_3$ concentrations for 2015-2017 within NAPAPCP and
2017-2019 within TYAPFAP, and the differentiated impacts of emission controls on
air quality were revealed for the two phases.

## 2. Methodology and data

### 2.1 Emission estimation

#### 2.1.1 Emission source classification

We applied a four-level framework of emission source categories for Jiangsu emission
inventory, based on a thorough investigation on the energy and industrial structures in
the province. The framework included six first-level categories this study, covering all
the social and economic sectors in Jiangsu: power sector, industry, transportation,
agriculture, residential, and biogenic source (for NMVOCs only). Moreover, the
framework contained fifty-five second-level categories based on facility/equipment
types and economical subsectors (see details in Table S1 in the Supplement), 240
third-level categories classified mainly by fuel, product, and material types, and a total
of 870 fourth-level categories including sources by combustion, manufacturing and
emission control technologies of emission facilities.
Compared to guidelines for development of national emission inventories (He et al.,
2018), forty-two new categories (third-level) were added in this study, contained
mainly in the second-level categories including metal products and the mechanical
equipment manufacturing industries, non-industrial solvent usage from ship fittings



and repairs, household appliances, and housing retrofitting emissions. Those
categories were identified as important sources of NMVOCs emissions in Jiangsu. In
particular, ship coating emissions, coming mainly from solvent usage during spraying,
cleaning and gluing in a wide range of procedures, could account for nearly 20% of
the solvent use emissions in the YRD region (Mo et al., 2021). Therefore, the updated
framework provided a more complete coverage of source categories, thus was able to
considerably reduce the bias of emission estimation due to missing potentially
important emitters.

**2.1.2 Emission estimation methods**

We applied the "bottom-up" methodology (i.e., the emissions were calculated at the
finest source level (e.g., facility level if data allowed) and then aggregated to upper
categories/regions) to develop the high-resolution emission inventory for Jiangsu (and
its 13 cities, as shown in Figure S1 in the Supplement) 2015-2019. As mentioned in
Introduction, we have conducted a series of studies and made substantial
improvements on the methodology of regional emission inventory development by
source category or species, compared to the ones at larger spatial scales. Here we
integrated those improvements as briefly described below, and additional further
details can be found in corresponding published articles.
**Power plant** We developed a method of examining, screening and applying online
measurement data from the continuous emission monitoring systems (CEMS, Zhang
et al., 2019b) to estimate the emissions at the power unit/plant level. For units without
CEMS data, we applied the average flue gas concentrations obtained from CEMS for
units with the same installed capacity. The emissions were calculated based on the
annual mean hourly flue gas concentration of air pollutant obtained from CEMS and
the theoretical annual flue gas volume of each unit/plant:

$$E_{i,j} = C_{i,j} \times AL_j \times V_m^0 \tag{1}$$

where $E$ is the emission of air pollutant; $i$, $j$ and $m$ represent the pollutant species,
individual plant/unit, and fuel type, respectively; $C$ is the annual average





concentration in the flue gas; $AL$ is the annual coal consumption, and $V^0$ is the
theoretical flue gas volume per unit of fuel consumption, which depends on the coal
type and can be calculated following the method in Zhao et al. (2010).
**Industrial plant** Emissions were principally calculated based on activity level data
(production output or energy consumption) and emission factor (emissions per unit of
activity level). For point sources with abundant information, we used a
procedure-based approach to calculate the emissions of pollutants (Zhao et al., 2017).
For example, we subdivided the iron and steel industry into sintering, pelletizing, iron
making, steel making, rolling steel, and coking. The activity data and emission factors
of each procedure were derived based on multiple information collected from
enterprise regular report, statistics, and/or on-site investigation at the facility level (see
Section 2.1.3). The emissions of air pollutants were calculated using Eq. (2):

$$E_i = \sum_{j,r} AL_{j,r} \times EF_{i,j,r} \times (1 - \eta_{i,j,r}) \tag{2}$$

where $r$ is the industrial procedure; $AL$ is the activity level; $EF$ is the unabated
emission factor; $\eta$ is the pollutant removal efficiency of end-of-pipe control
equipment.
**Petrochemical industry** Certain procedures in petrochemical industry have been
identified as the main contributors to AVOCs emissions from the sector. For example,
equipment leaks, storage tanks, and manufacturing lines were estimated to be
responsible for over 90% of the total emissions (Ke et al., 2020; Liu et al., 2020; Yen
and Horng, 2009). Through field measurements and in-depth analysis of different
emission calculation methods, Zhang et al. (2021a) suggested that procedure-based
method should provide better estimate of NMVOCs emissions for petroleum
industries than the commonly approach that applied a full emission factor for the
whole factory. In this study, therefore, we applied the procedure-based method for
four key procedures (manufacturing lines, storage tanks, equipment leaks, and
wastewater collection and treatment system), with best available information from
on-site surveys and regular enterprise reports.
**Agriculture** Agricultural $NH_3$ emissions can be significantly influenced by the
meteorological, soil environment, and farming manners, and thus are more difficult to





track compared to $SO_2$ and $NO_X$ that are largely from power and industrial plants. For
example, the relatively high temperature and top dressing fertilization conducted in
summer could significantly elevate the $NH_3$ volatilization for urea fertilizer use in
YRD. Our previous work (Zhao et al., 2020) quantified the effects of metrology, soil
property and various agricultural processes (e.g., fertilizer use and manure
management) on YRD $NH_3$ emissions for 2014. Here we expanded the research period
and obtained the agricultural $NH_3$ emission inventory for 2015-2019 in Jiangsu.
**Off-road transportation** We developed a novel method to estimate the emissions and
their spatiotemporal distribution for in-use agricultural machinery, by combining
satellite data, land and soil information, and in-house investigation (Zhang et al.,
2020). In particular, the machinery usage was determined based on the spatial
distribution, growing and rotation pattern of the crops. Moreover, twelve construction
and agricultural machines with different power capacity and emission grades (China
I-III) were selected and emission factors were measured under various working loads
(unpublished). In this work, we combined the method developed by Zhang et al.
(2020) and newly tested emission factors to estimate the emissions from off-road
machines in Jiangsu for multiple years.
**Biogenic source:** Located in the subtropics, Jiangsu has abundant broadleaf
vegetation, a main contributor to biogenic volatile organic compounds (BVOCs)
emissions. Our previous work (Wang et al., 2020b) evaluated the effect of land cover
data, emission factors and $O_3$ exposure on BVOCs emissions in YRD with the Model
of Emissions of Gases and Aerosols from Nature (MEGAN). Here we followed the
improved method by Wang et al. (2020b) and calculated BVOCs emissions with
integrated land cover information, local BVOCs emission factors, and influence of
actual $O_3$ stress in Jiangsu.
**Other sources** Emissions from on-road vehicles and residential sectors were
estimated following our previous work (Zhou et al., 2017; Zhao et al., 2021), with
updated activity levels and emission factors.
**NMVOCs speciation** We updated NMVOCs speciation by incorporating the local
source profiles from field measures (Zhao et al., 2017; Zhang et al., 2021a) and



massive literature reviews of previous studies (Mo et al., 2016; Li et al., 2014; Huang
et al., 2021; Wang et al., 2020a). Compared with the widely used SPECIATE 4.4
database (https://www.epa.gov/air-emissions-modeling/speciate, Hsu et al., 2018), we
included new source profiles from local measurements for production of sugar,
vegetable oil and beer, and refined the source profiles for the use of paints, inks,
coatings, dyes, dyestuffs and adhesives in manufacturing industry (Zhang et al.,
2021a), and selected production processes of chemical engineering (Zhao et al., 2017).
Moreover, we split the source profiles for some categories into finer ones, for example,
NMVOCs release in filling station into petrol and diesel release, metal surface
treatment into water-based and solvent-based paints, and ink printing into offset,
gravure and letterpress printing. Those efforts made the NMVOCs speciation more
representative for local emission sources (Zhang et al., 2021a).
**2.1.3 Data compilation, investigation and incorporation**
In this study, we compiled, investigated and incorporated most available information
on emission sources to improve the completeness, representativeness and reliability of
provincial emission inventory. In particular, we collected officially reported
Environmental Statistics Database (ESD, 2015-2019) and the Second National
Pollution Source Census (SNPSC, 2017) for information of stationary sources (mostly
power and industrial ones). Both of them contained basic information on location, raw
material and energy consumption, product output, and manufacturing and emission
control technologies. The former was routinely reported for relatively big point
sources every year, but some information could be outdated or inaccurate attributed to
insufficient on-site inspection. Through wide on-site surveys, in contrast, the latter
included much more plants, and provided or corrected crucial information at facility
level, such as removal efficiency of air pollutant control devices (APCD). However,
the database was developed for 2017 and could not track the changes for recent years.
Therefore, we further applied an internal database from the Air Pollution Source
Emission    Inventory    Compilation    and    Analysis    System    (APSEICAS,



http://123.127.175.61:31000), which was developed by Jiangsu Provincial Academy
of Environmental Sciences. Following the principal of SNPSC, the information of
APSEICAS has been collected and dynamically updated since 2018, based mainly on
in-depth investigation for individual enterprises conducted jointly by themselves and
local environmental administrators. We made cross validation and necessary revision
according to above-mentioned three databases, to ensure the accuracy of information
as much as possible.
As a result, we obtained sufficient numbers of point sources with satisfying
facility-level information for provincial-level emission inventory development
(57,457, 32,324 and 48,826 for 2017, 2018, and 2019, respectively). The shares of
coal consumption by those sources to the total ranged 90-94% for the three years. The
high proportions of point sources could effectively reduce the uncertainty in
estimation and spatial allocation of air pollutant emissions. For the remaining
industrial sources, the emissions were calculated with the average emission factor by
city and sector, and were spatially allocated according to the distribution of local
industrial parks and GDP.
Other information including area industrial sources, transportation, agricultural, and
residential sources were taken from economical and energy statistical yearbooks at
city level. Activity data that were not recorded (e.g., civil solvent usage, catering, and
biomass burning) were indirectly estimated from relevant statistics, including
population, building area, and crop yields.

## 2.2 Analysis of emission change

In this study, we summarized 13 major control measures adopted between 2015 and
2019, based on NAPAPCP, TYAPFAP and relative action plans promulgated by the
Jiangsu government (Figure S2 in the Supplement). Those included Ultra-low
emission retrofit of coal-fired power plants, Extensive management of coal-fired
boilers, Upgradation and renovation of non-electrical industry, Phasing out outdated
industrial capacities, Promoting clean energy use, Phasing out small polluting





factories, Construction of port shore power, Comprehensive treatment of mobile
source pollution, VOCs emission control in key sectors, Application of leak detection
and repair (LDAR), Oil and gas recovery, Replacement with low-VOC paints, Control
of non-point pollution. We applied the method by Zhang et al. (2019a) to quantify the
benefits of those air clean actions on emission abatement. Briefly, the emission
reduction resulting from implementation of a specific measure was estimated by
changing the parameters of emission calculation associated with the measure within
the concerned period, and keeping other parameters constant (same as initial year).
The emission reduction from each measure was then estimated for 2015-2017 and
2017-2019. The provincial-level emission inventory developed in Section 2.1 was
adopted as the baseline of the emission estimates. It was worth noting that the
aggregated emission reduction from all the measures did not equal to the net reduction,
as the factors leading to emission growth were not counted in this analysis.

## 2.3 Air quality modeling

### 2.3.1 Model configurations

To evaluate the provincial-level emission inventory, we used the Community
Multiscale Air Quality (CMAQ v5.1) model developed by US Environmental
Protection Agency (USEPA), to simulate the $PM_{2.5}$ and $O_3$ concentrations in Jiangsu.
Four months (January, April, July, and October) of each year between 2015 and 2019
were selected as the simulation periods, with a spin-up time of 7 days for each month
to reduce the impact of the initial condition on the simulation. As shown in Figure S1,
three nested domains (D1, D2, and D3) were applied with the horizontal resolutions at
27, 9, and 3 km, respectively, and the most inner D3 covered Jiangsu and parts of the
YRD region including Shanghai, northern Zhejiang, and eastern Anhui. MEIC was
applied for D1, D2, and the regions out of Jiangsu in D3, and the provincial-level
emission inventory was applied for Jiangsu in D3. The Carbon Bond Mechanism
(CB05) and AERO5 mechanisms were used for the gas-phase chemistry and aerosol
module, respectively.





The meteorological field for the CMAQ model was obtained from the Weather
Research and Forecasting model (WRF v3.4). Meteorological initial and boundary
conditions were obtained from the National Centers for Environmental Prediction
(NCEP) datasets. Ground observations at 3-h intervals were downloaded from
National Climatic Data Center (NCDC). Statistical indicators including bias, index of
agreement (IOA), and root mean squared error (RMSE) were used to evaluate the
WRF performance (Yang et al., 2021a). The discrepancies between simulations and
ground observations were within an acceptable range (Table S2 in the Supplement).
In order to evaluate the model performance of CMAQ, we collected ground
observation data of hourly $PM_{2.5}$ and $O_3$ concentrations at the 110 state-operating air
quality monitoring stations within Jiangsu (https://data.epmap.org/page/index, see the
station locations in Figure S1). Correlation coefficients (R), normalized mean bias
(NMB) and normalized mean errors (NME) between observation and simulation for
each month were calculated to evaluate the performance of CMAQ modeling:
$$NMB = \sum_{p=1}^{n}(S_p - O_p)/\sum_{p=1}^{n} O_p \times 100\% \qquad (3)$$
$$NME = \sum_{p=1}^{n}|S_p - O_p|/\sum_{p=1}^{n} O_p \times 100\% \qquad (4)$$
where $S$ and $O$ are the simulated and observed concentration of air pollutant,
respectively, and $p$ indicates the individual year (n=5 in this study).
We further compared the modeling performance using provincial-level emission
inventory in D3 with that using MEIC in D2. Zheng et al. (2017) suggested a much
larger bias for high-resolution simulation (additional 8-73% at 4 km) than that at
coarser resolution (3-13% for 36 km) when MEIC was applied in predicting surface
concentrations of different air pollutants. To avoid expanded modeling bias, therefore,
we did not directly downscale MEIC into the entire D3.
**2.3.2 Emission and meteorological factors affecting the variation of $PM_{2.5}$ and $O_3$**
We set up different scenarios to assess the impacts of emission and meteorological
changes on the interannual variations of $PM_{2.5}$ and $O_3$ concentrations, and to reveal
their varying contributions for different periods. The baseline represented the





simulation for 2015, 2017, and 2019 with the emission inventory and meteorological
fields for corresponding year. The meteorological variation scenario (VMET) used the
varying meteorological fields for the three years but fixed the emission input at the
2017 level, and was thus able to quantify the impact of changing meteorological
conditions on $PM_{2.5}$ and $O_3$ concentrations. For example, the difference between 2015
and 2017 in VMET indicated the contribution of changing meteorology to variation of
air pollutant concentration (same for the period 2017-2019). Similarly, the emission
variation scenario (VEMIS) used the varying emission inventory for the three years
but fixed meteorological fields at the 2017 level, and was thus able to quantify the
impact of changing emissions on $PM_{2.5}$ and $O_3$ concentrations. For example, the
difference between 2015 and 2017 in VEMIS indicated the contribution of changing
emissions to variation of air pollutant concentration (same for the period 2017-2019).
The contributions between 2015 and 2017, and those between 2017 and 2019, could
then be compared to evaluate the effectiveness of emission control on air quality for
the two periods. Notably the emission change in the modeling scenario referred to that
for entire D3, thus the contribution of emission control to the changing air quality
included both from Jiangsu and nearby regions. Given the clearly larger emission
intensity for the former compared to the latter (An et al., 2021), the contribution of
local emissions was expected to be more important on the air quality than regional
transport.

## 3. Results and discussions

### 3.1 Air pollutant emissions by sector and region

#### 3.1.1 Anthropogenic emission changes by sector

From 2015 to 2019, the total emissions of anthropogenic $SO_2$, $NO_X$, AVOCs, $NH_3$,
CO, $PM_{10}$, $PM_{2.5}$, BC, and OC in Jiangsu were estimated to decline 53%, 20%, 6%,
10%, 7%, 21%, 16%, 6% and 18%, down to 296, 1122, 1271, 422, 7163, 565, 411, 32,
and 36 Gg in 2019, respectively (Table S3 in the Supplement). On top of $SO_2$ and



$NO_X$, NMVOCs has been incorporated into national economic and social
development plans with emission reduction targets in China since 2015, because of its
harmful impact on human health and increasingly important role on triggering $O_3$
formation. The central government required the total national emissions of $SO_2$, $NO_X$,
and NMVOCs to be cut by 15%, 15%, and 10% during the 13th Five-Year Plan period
(2015-2020), respectively (Zhang et al., 2022). Our estimates show that the actual $SO_2$
and $NO_X$ emission reductions were larger than planned in Jiangsu, due to the
implementation of stringent pollution control measures. However, AVOCs emissions
did not decline considerably within the research period, resulting from less
penetration of efficient APCD, and more fugitive leakage that were difficult to capture.
Relatively small reductions were also found for BC and CO, which are closely
associated with incomplete combustion of small-scale sources and vehicles. The lack
of APCD and growth of vehicle use were expected to offset the benefits of emission
controls for other sectors. As shown in Figure 1, the GDP and vehicle population grew
40% and 24%, respectively, while coal consumption declined slightly during
2015-2019. Along with stringent emission reduction actions, the provincial emissions
of $SO_2$, $NO_X$ and $PM_{2.5}$ were clearly decoupling from those economical and energy
factors, while CO was still strongly influenced by the change of coal consumption.
We present the sectoral contribution to anthropogenic emissions and their interannual
changes in Figure 2 and Figure 3, respectively. Industrial sector was identify as the
major contributor to $SO_2$, CO, AVOCs, $PM_{10}$, and $PM_{2.5}$ emissions, accounting
averagely for 50%, 62%, 64%, 68%, and 61% of them during 2015-2019, respectively
(Figure 2a, c, d, f and g). The sector was found to drive the reductions in emissions of
$SO_2$, $NO_X$, CO, $PM_{10}$, $PM_{2.5}$ and BC. In particular, the benefit of emission controls on
industrial sector after 2017 was found to clearly elevated and to surpass that of power
sector for $SO_2$, $NO_X$, $PM_{10}$ and $PM_{2.5}$ (Figure 3a, b, f and g).
The power sector, accounting for more than half of provincial coal burning though,
was not the most important contributor to the emissions of any pollutant (Figure 2).
Upgrading the units with advanced APCDs, phasing-out outdated boilers, and
retrofitting for ultra-low emission requirement significantly reduced $SO_2$, $NO_X$, and



particulate emissions from the power sector (Liu et al., 2015; Zhang et al., 2021b).
With the completion of the ultra-low emission retrofit in 2017, the declines of
emissions for most species slowed down for the power sector (Figure 3). The results
indicated that the potential for further emission abatement from end-of-pipe controls
has been very limited for the sector, unless an energy transition with less coal
consumption is sustainably undertaken in Jiangsu.
The transportation sector averagely accounted for 51%, 17%, 14% and 42% of $NO_X$,
CO, AVOCs and BC emissions, respectively (Figure 2b, c, d, and h). The growth of
vehicle population resulted in a 38% increase in the annual $NO_X$ emissions from
transportation from 2015 to 2019, faster than that of any other sector (Figure 3b).
Similarly, a 20% and 25% increase were found for transportation CO and BC
emissions (Figure 3c and h), respectively. Therefore, the rapid development of
transportation in economically developed Jiangsu has expanded its contribution to air
pollutant emissions for those species, particularly after the emissions from large
power and industrial plants have been effectively curbed. However, implementation of
China  V  emission  standard  (equal  to  Euro  V,
https://publications.jrc.ec.europa.eu/repository/handle/JRC102115) for motor vehicles
since 2018 effectively slowed down the growth of transportation $NO_X$ emissions: The
annual growth rate was estimated to decrease from 12% for 2015-2017 to 5% in
2018-2019. Meanwhile, a downward trend was also found for transportation AVOCs
emissions since 2018 (Figure 3d). Those results show that emission controls for
transportation could be crucial for limiting the key precursors of ozone production
(Geng et al., 2021; Zhang et al.,2019a).
The residential sector was the most important source of OC, contributing averagely 68%
to total emissions within 2015-2019 (Figure 2i), and was the second most important
source of $PM_{10}$ (18%, Figure 2f) and $PM_{2.5}$ (24%, Figure 2g). It dominated the
abatement of OC emissions, attributed to the reduced bulk coal and straw burning
(Figure 3i). The agricultural sector dominated $NH_3$ emissions (91%, Figure 2e), and
the small decline resulted mainly from the reduced use of nitrogen fertilizer (13%)
from 2015 to 2019 (Figure 3e).



### 3.1.2 City-level emissions and spatial distribution

Figure 4 shows the average annual emissions of $SO_2$, $NO_X$, AVOCs, $NH_3$, $PM_{2.5}$ for
the five years by city. Clearly larger emissions of most species were found in southern
Jiangsu cities (see the city definitions in Figure S1) with more developed industrial
economy and transportation (Figure 4a-e, see the detailed emission data by year and
city in Table S4 in the Supplement). The $SO_2$ emissions per unit area were calculated
at 7.7, 3.3, and 2.4 ton·$km^{-2}$ for southern, central and northern cities, respectively. The
analogous numbers were 23.0, 11.7, and 8.1 ton·$km^{-2}$ for $NO_X$, 22.5, 13.2, and 8.1
ton·$km^{-2}$ for AVOCs, and 7.3, 5.2, and 2.9 ton·$km^{-2}$ for $PM_{2.5}$, respectively. As shown
in Figure S3 in the Supplement, the regions along the Yangtze River were of largest
densities of power and industrial plants. In contrast, higher $NH_3$ emissions were found
for central and northern cities with abundant agricultural activities (Figure 4). Figure
S4 in the Supplement illustrates the spatial distributions of emissions for selected
species for 2019, at a horizontal resolution of 3km. Besides industrial sources, the
spatial patterns of $NO_X$, BC, CO and AVOCs were also influenced by the road net,
suggesting the role of heavy traffic on emissions. Particulate matter emissions were
mainly distributed in urban industrial regions, while OC was more found in broader
central and northern areas, attributed partly to the contribution from residential biofuel
use.
As shown in Figure 4, the emission fractions of southern cities decreased from 2015
to 2019 except for AVOCs and $NH_3$, indicating more benefits of stringent measures on
emission controls for relatively developed regions. Faster declines in annual $SO_2$,
$NO_X$ and $PM_{2.5}$ emissions for southern cities (59%, 23%, and 24% from 2015 to 2019,
respectively) were estimated than northern cities (53%, 18%, and 8%, respectively).
In contrast, AVOCs emissions were estimated to increase by 10% in southern cities
while decrease by 27% in northern cities.
Figure 5 illustrates the changes in spatial distribution of major pollutant emissions
from 2015 to 2019 in Jiangsu. It can be found that the areas with large emission
reduction for $SO_2$, $NO_X$, and $PM_{2.5}$ were consistent with the locations of super



emitters of corresponding species (Figure 5a-c). Facing bigger challenges in air
quality improvement, more efforts have been made on the emission controls of
large-scale power and industrial enterprises in the economically developed southern
Jiangsu, leading to greater emission reduction compared to less developed northern
Jiangsu. Opposite pattern in spatial variation of emissions was found for AVOCs
(Figure 5d). There was a big development of industrial parks for chemical engineering
along the riverside of Yangtze River in the cities of Suzhou, Nantong, and Wuxi in
southern Jiangsu. The elevated solvent use and output of chemical products of those
large-scale enterprises resulted in the growth of AVOCs emissions. In northern
Jiangsu, in contrast, small-scale chemical plants have been gradually closed, and the
emissions were thus effectively reduced. There is a thus great need for substantial
improvement of emission controls for the key regions and sectors for further
abatement of AVOCs emissions.

**3.1.3 Enhanced contribution of biogenic sources to total NMVOCs**

Table 1 summarizes AVOCs and BVOCs emissions by month and year. Different from
AVOCs that decreased slowly but continuously from 2015 to 2019, a clearly growth
of annual BVOCs emissions was estimated between 2015 and 2017, followed by a
slight reduction till 2019. The peak annual BVOCs emissions reached 213 Gg in 2017.
The interannual variation of BVOCs was mainly associated to that of temperature and
short-wave radiation (Wang et al., 2020b). Influenced by meteorological conditions
and vegetation growing season, BVOCs emissions were most abundant in July, less in
April and October and almost zero in January. Within the province, there existed a
general increasing gradient from southeast to northwest in BVOCs emissions (Figure
S5 in the Supplement). The rapid development of industrial economy in southern
Jiangsu has led to the expansion of urban centers and less vegetation cover, which
limited the BVOCs emissions.
We calculated the ratio of BVOCs to AVOCs emissions by month and year (Table 1).
Dependent on the emission trends of both BVOCs and AVOCs, the annual ratio





gradually increased from 11 in 2015 to 16 in 2017, and stayed above 15 afterwards.
There is also a clear seasonal difference in the ratio, with the averages for the five
years estimated at 0%, 8%, 52%, and 3% for January, April, July and October,
respectively. Since 2016, the ratio of BVOCs to AVOCs emissions exceeded 50% in
July, indicating that the $O_3$ pollution in summer could be increasingly influenced by
BVOCs. Regarding the spatial pattern, larger ratios were commonly found in northern
Jiangsu, with a modest growth for recent years (Figure 6). Moreover, greater growth
of the ratio was found in part of southern Jiangsu where AVOCs emissions were
rapidly declining (e.g., Nanjing and Zhenjiang). The evolution indicated that biogenic
sources gradually became more influential in $O_3$ production even for some regions
with developed industrial economy, along with controls of anthropogenic emissions.
Due to the relatively high level of ambient $NO_2$ from anthropogenic emissions, a
broad areas of Jiangsu were identified with a mixed or VOC-limited regime in terms
of $O_3$ formation (Jin and Holloway, 2015), indicating the impacts of NMVOCs
(including BVOCs) on the ambient $O_3$ concentration. In the future, the BVOCs
emissions may further increase with the elevated temperature, improved afforestation
and vegetation protection, and they will probably play a more important role on
summer $O_3$ pollution once the controls of AVOCs emissions are pushed forward (Ren
et al., 2017; Gao et al., 2022a).

**3.2 Influence of different data and methods on emission estimates**


**3.2.1 Assessment of interannual variability**


Figure 7 compares the interannual trends of $SO_2$ and $NO_X$ emissions estimated in this
study with those in available global (EDGAR) and national emission inventories
(MEIC), as well as those of annual averages of ambient concentrations for
corresponding species collected from the state-operating observation sites in Jiangsu.
Significantly different from other inventories, the global emission inventory EDGAR
could not reflect the rapid decline of $SO_2$ and $NO_X$ emissions of Jiangsu for recent
years. It was probably due to the lack of information on the gradually enhanced



penetrations and removal efficiencies of APCDs use in power and industrial sectors in
EDGAR.
Both MEIC and our provincial inventory show the continuous declines in SO₂ and
NO$_X$ emissions for Jiangsu from 2015 to 2019, which could be partly confirmed by
the ground observation. In general quite similar trends were found for the two
inventories, suggesting similar estimations in the interannual variation of total
emissions at the national and provincial scales. However, there existed some
discrepancies between the two. Compared to MEIC, as shown in Figure 7a, a slower
decline in SO₂ emissions between 2015 and 2017 was estimated by our provincial
inventory, but a faster one between 2017 and 2019. In other words, MEIC was more
optimistic in emission abatement for earlier years. The ultra-low emission retrofit on
power sector started from 2015 in Jiangsu, which was expected to largely reduce the
emissions of coal-fired plants to the level of gas-fired ones. Through investigations
and examinations of information on APCD operations for individual sources, we
cautiously speculated that the benefit of the retrofit might not be as large as expected
at the initial stage. This could be partly supported by the correspondence between
online monitoring of SO₂ emissions for individual power plants and satellite-derived
SO₂ columns around them when the ultra-low emission retrofit was required (Karplus
et al., 2018). From 2017 to 2019, we were more optimistic on the emission reduction,
attributed partly to larger benefit of emission controls on non-electric industries.
Similar case with less discrepancy could also be found for NO$_X$ emission (Figure 7b).
**3.2.2 Comparisons with previous studies**
To further evaluate the influence of data and methods on emission estimation, we
compared our provincial-level emission inventory with previous studies on emissions
in Jiangsu in terms of the total and sectoral emissions through examinations of activity
data, emission factor, removal efficiency and other parameters.
Table 2 compares our emission estimates, by year and species, with available
continental (REAS, Kurokawa et al., 2020), national (MEIC), and regional emissions





inventories (Li et al., 2018; Sun et al.,2018; Zhang et al., 2017b; Simayi et al., 2019;
An et al., 2021; Gao et al. 2022b). In particular, we stressed the differences in
emissions by sector among our study, MEIC and An et al. (2021) for 2017 as an
example (Figure 8).
The annual $SO_2$ emissions in our provincial inventory were close to those in REAS
(2015) and MEIC for most years, but much smaller than those reported by Sun et al.
(2018) and Li et al. (2018). The emissions in this work were 32% higher than the
MEIC for 2017, with the biggest difference (62% higher in this work) for power
sector (Figure 8). It resulted mainly from the discrepancies in the penetration and $SO_2$
removal efficiency of flue gas desulfurization (FGD) systems applied in the two
emission inventories. For example, Zhang et al. (2019a) assumed that the penetration
rate of FGD in the coal-fired power sector reached 99.6% in 2017, with the removal
efficiency estimated at 95%. According to our unit-based investigation, the removal
efficiencies in the power sector were typically less than 92%, owing to the aging
devices, low flue gas temperature and other reasons. The main differences between
this work and the YRD emission inventory by An et al. (2021) existed in the industrial
sector, attributed partly to insufficient consideration of the comprehensive emission
control regulations of coal-fired boilers in Jiangsu in the past few years in An et al.

618  (2021).

The estimates of $NO_X$ emissions from MEIC and Sun et al. (2018) were 14-26%
higher than ours. The major difference between MEIC and our provincial inventory
existed in the power and industrial sector, and the total emissions in the former were
56% larger than the latter (Figure 8). For example, the emission factors for coal-fired
power plants in this study were derived from CEMS (0.03-2.8 $g \cdot kg^{-1}$ coal),
significantly smaller than those from applied in MEIC and other research (2.88-8.12
$g \cdot kg^{-1}$ coal, Zhang et al., 2021b). Similarly, the smaller emission factors for industrial
boilers derived based on on-site investigations were commonly smaller than previous
studies, leading to an estimation 45% smaller than MEIC for industrial sector in 2017.
Correspondingly, some modeling and satellite studies suggested that the $NO_X$
emissions in previous studies were overestimated partly due to less consideration of





improvement in NOx control measures for coal burning sources (Zhao et al., 2018;
Sha et al., 2019).
As mentioned in Section 2.1.2, AVOCs emissions for certain industrial sources in this
study were estimated with a procedure-based approach, which took the removal
efficiencies of different technologies into account (Zhang et al., 2021a). Therefore, the
annual AVOCs emissions in the provincial inventory were commonly much smaller
than others. Without sufficient the local information, for example, Simaya et al. (2019)
applied the national average removal efficiencies of AVOCs in furniture
manufacturing, automotive manufacturing and textile dyeing industries at 18%, 28%,
and 30%, clearly lower than 21%, 42%, and 43% in our inventory, respectively. As a
result, the AVOCs emissions from industrial source in the former were 45% higher
than the latter.
$NH_3$ emissions in the provincial emission inventories were commonly smaller than
others. In particular, the estimate was less than half of that by An et al. (2021) for
2017 (Figure 8). The big difference resulted mainly from the methodologies. As
indicated by our previous study (Zhao et al., 2020), the method characterizing
agricultural processes usually provided smaller emission estimates than that using the
constant emission factors. The former detected the emission variation by season and
region, and was more supportive for air quality modeling with better agreement with
ground and satellite observation.
For PM emissions, our estimates were larger than MEIC, Gao et al. (2022b), and An
et al. (2021). The discrepancies resulted mainly from the inconsistent penetration rates
and removal efficiencies of dust collectors determined at national level and from
on-site surveys at provincial level. Taking cement as an example, all the plants were
assumed to be installed with dust collectors, and the national average removal
efficiency was determined at 99.3% in MEIC (Zhang et al., 2019a), clearly larger than
that in Jiangsu from plant-by-plant surveys (93%). Thu the $PM_{10}$ and $PM_{2.5}$ emissions
from the industrial sector in this study were 197 and 113 Gg higher than MEIC for
2017 (Figure 8).



### 3.3 Analysis of driving force of emission change from 2015 to 2019

The actual reductions of annual $SO_2$, $NO_X$, AVOCs, $NH_3$, and $PM_{2.5}$ emissions were estimated at 331, 289, 77, 46, and 80 Gg from 2015 to 2019, respectively in our provincial emission inventory. We analyzed the emission abatement and its driving forces for two periods, 2015-2017 and 2017-2019, to represent the different influences of individual measures on emissions for NAPAPCP and TYAPFAP. As shown in Figure S6 in the Supplement, the actual emission reductions of $SO_2$ and $NH_3$ during 2015-2017 (211 and 34 Gg respectively) exceeded those during 2017-2019 (120 and 12 Gg, respectively). As the retrofit of ultra-low emission technologies for the power sector and the modification of large-scale intensive management of livestock farming in Jiangsu were basically completed between 2015 and 2017. The reductions of annual $NO_X$, AVOCs, and $PM_{2.5}$ emissions during 2017-2019 were significantly larger (209, 72, and 57 Gg, respectively) than those during 2015-2017 (80, 5, and 23 Gg, respectively), implying bigger benefits of TYAPFAP on emission controls of those species.

Figure 9 summarizes the effect of individual measures on net emission reduction for the two periods. The ultra-low emission retrofit of coal-fired power plants was identified to be the most important driving factor for the reductions of $SO_2$ and $NO_X$ emissions during 2015-2017, responsible for 38% and 43% of the abatement for the two species, respectively. By the end of 2017, more than 95% of the coal-fired power plants in Jiangsu were equipped with FGD and selective catalytic/non-catalytic reduction (SCR/SNCR), and 91% of coal-fired power generation capacity met the ultra-low emission standards (35, 50 and 10 mg·m$^{-3}$ for $SO_2$, $NO_X$ and PM concentration in the flue gas, respectively; Zhang et al., 2019a). Through the information cross check and incorporation based on different emission source databases as mentioned in Section 2.1.3, the average removal efficiencies of $SO_2$ and $NO_X$ in the coal-fired power plants were estimated to increase from 89% and 50% in 2015 to 94% and 63% in 2017, respectively.

The extensive management of coal-fired boilers was the second most important driver



688 for $SO_2$ and $NO_X$ reduction and the most important driver for $PM_{2.5}$, contributing to

689 24%, 20% and 37% of the emission reductions for corresponding species, respectively.

690 The main actions included the elimination of 100 MW of coal-fired power generation

691 capacity and the enhanced penetrations of $SO_2$ and particulate control devices on large

692 coal-fired industrial boilers since the improved enforcement of the latest emission

693 standard (GB 13271–2014).

694 The upgradation and renovation of non-electrical industry contributed 18%, 15%, and

695 28% to the emission reductions for $SO_2$, $NO_X$, and $PM_{2.5}$, respectively. Till 2017,

696 more than 80% of steel-sintering machines and cement kilns were equipped with FGD

697 and SCR/SNCR systems. The average removal efficiency in the steel and cement

698 production increased from 48% and 43% in 2015 to 60% and 57% in 2017 for $SO_2$,

699 and from 45% and 38% in 2015 to 54% and 40% in 2017 for $NO_X$, respectively (as

700 shown in Figure S7 in the Supplement).

701 Phasing out outdated capacities in key industries including crude steel (8 million tons),

702 cement (9 million tons), flat glass (3 million weight-boxes), and other

703 energy-inefficient production capacity contributed 11%, 6%, and 11% to the emission

704 reductions of corresponding species, respectively. Given their relatively small

705 proportions to total emissions, the contributions of other emission reduction measures

706 were less than 10%, such as promoting clean energy, phasing out small and polluting

707 factories, and the construction of port shore power.

708 The driving forces of emission abatement have been changing since implementation

709 of TYAPFAP. The potential for further reduction of $SO_2$ and $NO_X$ emissions were

710 significantly narrowed through the end-of-pipe treatment in the power sector, and the

711 ultra-emission retrofit on the sector was of very limited influence on the emissions

712 during 2017-2019. Measures on the non-electric sector brought greater benefits on

713 emission reduction. Extensive management of coal-fired boilers and upgradation and

714 renovation of non-electrical industry maintained as the most important driving factors

715 for the reduction of $SO_2$, $NO_X$, and $PM_{2.5}$ emissions (33%, 20%, and 26% for the

716 former and 28%, 29% and 33% for the latter, respectively). After 2017, small coal





boilers (≤30 MW) were continuously shut down and remaining larger ones (≥60
MW) were all retrofitted with ultra-low emission technology. Through the ultra-low
emission retrofit, the average removal efficiencies of $NO_X$ in the steel and cement
production increased from 54% and 40% in 2017 to 70% and 61% in 2019,
respectively.
AVOCs emission reduction resulted mainly from implementation of controls on the
key sectors, which accounted for 63% and 34% of the reduced emissions for
2015-2017 and 2017-2019, respectively. Besides, application of LDAR was the
second most important measure for 2015-2017, with the contribution to emission
reduction reaching 23%. The results also showed that AVOCs emission reductions
from all the concerned measures in 2017-2019 (152Gg) were higher than those in
2015-2017 (116 Gg). Although more abatement in total AVOCs emissions was found
for 2017-2019 (Figure S6), the contributions of above-mentioned two measures
reduced clearly in the period. Some other measures were identified to be important
drivers of emission reduction, including control on mobile sources (e.g.,
implementation of the China V emission standard for on-road vehicles) and
replacement with low-VOCs paints. In our recent studies, we evaluated the average
removal efficiency of AVOCs in industrial sector was less than 30% (Zhang et al.,
2021a), and organic solvents with low-VOCs content accounted for less than 30% of
total solvent use (Wu et al., 2022). Therefore, there would still be great potential for
further reduction of AVOCs emissions through improvement on the end-of-pipe
emission controls and use of cleaner solvents.
In a summary, expanding the end-of-pipe treatment (e.g., the ultra-low emission
retrofit) from power to non-electricity industry and phasing out the outdated industrial
capacities have been driving the declines of emissions for most species. Along with
the limited potential for current measures, more substantial improvement of energy
and industrial structures could be the option for further emission reduction in the
future.



### 3.4 Effectiveness of emission controls on the changing air quality

#### 3.4.1 Simulation of the $O_3$ and $PM_{2.5}$ concentration

The CMAQ model performance was evaluated with available ground observation. The observed concentrations of $PM_{2.5}$ (hourly) and $O_3$ (the maximum daily 8-h average, MDA8) were compared with the simulations using the provincial emission inventory and MEIC for the selected four months for 2015-2019, as summarized in Table S5 and Table S6 in the Supplement. Overall, the simulation with the provincial inventory shows acceptable agreement with the observations, with the annual means of NMB and NME ranging -21% – 2% and 43% –52% for $PM_{2.5}$, and -26% – -14% and 30% – 41% for $O_3$. The analogous numbers for MEIC were -23% – -5% and 47% – 53% for $PM_{2.5}$, and -26% – -6% and 33% – 46% for $O_3$, respectively. Most of the NMB and NME were within the proposed criteria (-30%≤NMB≤30% and NME≤50%, Emery et al., 2017). Better performance was achieved using the provincial inventory, implying the benefit of application of refined emission data on high-resolution air quality simulation.

Figure 10 compares the observed and simulated (with the provincial inventory) interannual trends in $PM_{2.5}$ and MDA8 $O_3$ concentrations from 2015 to 2019 (see the simulated spatiotemporal evolution in Figures S8 and S9 in the Supplement). Satisfying correlations between observed and simulated concentrations were found for both $PM_{2.5}$ and MDA8 $O_3$, with the squares of correlation coefficients ($R^2$) estimated at 0.81 and 0.86 within the research period, respectively. The good agreement suggested the simulation with high-resolution emission inventory was able to well capture the interannual changes in air quality at the provincial scale.

Both observation and simulation indicated a declining trend of $PM_{2.5}$ concentrations, with the annual decreasing rates estimated at -5.4 and -4.2 $\mu g \cdot m^{-3} \cdot yr^{-1}$, respectively (Figure 10a). The decline reflected the benefit of improved implementation of emission control actions as well as the influence of meteorological condition change. In general, higher concentrations were found in summer and lower in winter. A





rebound in PM$_{2.5}$ level was notably found in winter after 2017, attributed possibly to
the unfavorable meteorological conditions for recent years. In contrast to PM$_{2.5}$,
MDA8 O$_3$ was clearly elevated from 2015 to 2019, with the annual growth rates
estimated at 4.6 and 7.3 μg·m$^{-3}$·yr$^{-1}$, by observation and simulation (Figure 10b).
Higher levels were found in spring and summer and lower in autumn and winter.
Besides the impact of emission change, the O$_3$ concentrations can be greatly
influenced by the varying meteorological factors such as the decreased relative
humidity and wind speed for recent years in YRD region (Gao et al., 2021; Dang et al.,
2021). In addition, the recent declining PM$_{2.5}$ concentration in eastern China reduced
the heterogeneous absorption of hydroperoxyl (HO$_2$) radicals by aerosols and thereby
enhanced O$_3$ concentration (Li et al., 2019). The complicated impacts of various
factors on air quality triggered the separation of emission and meteorological
contributions to the changing PM$_{2.5}$ and O$_3$ levels in Section 3.4.2.
The common underestimation of O$_3$ should be stressed, partly resulting from the bias
in the estimation of precursor emissions. In this study, the enhanced penetrations
and/or removal efficiencies of NO$_X$ control devices might not be fully considered in
the emission inventory development, in particular for the non-electric industry,
leading to possible overestimation of NO$_X$ emissions. Moreover, underestimation of
AVOCs emissions could exist, due to incomplete counting of emission sources,
particularly for the uncontrolled fugitive leakage. As most of Jiangsu was identified as
a VOC-limited region for O$_3$ formation (Wang et al., 2020b; Yang et al., 2021b), the
overestimation of NO$_X$ and underestimation of AVOCs could resulted in
underestimation in O$_3$ concentration with air quality modeling. Furthermore, a larger
underestimation in O$_3$ was revealed before 2017 (Figure 8b), attributed partly to less
data support on the emission sources and thereby less reliability in the emission
inventory, compared with more recent years.
**3.4.2 Anthropogenic and meteorological contribution to O$_3$ and PM$_{2.5}$ variation**
Figure 11 explores the effects of the changing anthropogenic emissions (VEIMS) and



meteorology (VMET) on PM$_{2.5}$ and MDA8 O$_3$ levels in 2015-2017 and 2017-2019. In
the baseline that contained the interannual changes of both factors, the
provincial-level PM$_{2.5}$ concentration was simulated to decrease by 4.1 μg·m$^{-3}$ in
2015-2017 and 1.7 μg·m$^{-3}$ in 2017-2019, and MDA8 O$_3$ increase by 17.0 μg·m$^{-3}$ in
2015-2017 and 3.2 μg·m$^{-3}$ in 2017-2019. Therefore, smaller variations were found for
more recent years for both species. Due to nonlinearity effect of the chemistry
transport modeling, the air quality changes in baseline did not equal to the aggregated
contributions in VMET and VEMIS.
As shown in Figure 11a, similar patterns of driving factor contributions to PM$_{2.5}$ were
found between 2015-2017 and 2017-2019. While meteorological conditions
consistently promoted the formation of PM$_{2.5}$, the continuous abatement of
anthropogenic emissions completely offset the adverse meteorological effects and
contributed significantly to the declines in PM$_{2.5}$ concentrations. Although less
reduction in PM$_{2.5}$ concentration was found for 2017-2019 due mainly to the worsened
meteorology, emission abatement were estimated to play a greater role on reducing
PM$_{2.5}$ concentration (5.5 μg·m$^{-3}$ in VEMIS) compared to 2015-2017 (4.3 μg·m$^{-3}$),
implying the bigger effectiveness of recent emission control actions on PM$_{2.5}$
pollution alleviation.
The O$_3$ case is different (Figure 11b). Both the changing emissions and meteorology
favored MDA8 O$_3$ increase for 2015-2017, consistent with previous studies (Wang et
al., 2019; Dang et al., 2021). The contribution of meteorology was estimated at 11.9
μg·m$^{-3}$ (VMET), larger than that of emissions at 4.9 μg·m$^{-3}$ (VEMIS). As shown in
Figure S6, the abatement of annual NO$_X$ emissions in Jiangsu was estimated at 104
Gg, while very limited reduction was achieved in AVOCs emissions. Declining NOx
emissions thus elevated O$_3$ formation under the VOC-limited conditions particularly
in urban areas in Jiangsu.
During 2017-2019, the meteorological condition played a more important role on the
O$_3$ growth (14.3 μg·m$^{-3}$), attributed mainly to the decreased relative humidity and
wind speed for recent years (Table S2). In contrast, the changing emissions were
estimated to restrain the O$_3$ growth by 3.1 μg·m$^{-3}$, implying the effectiveness of



continuous emission controls on $O_3$ pollution alleviation. As shown in Figure S6, a
much larger reduction in AVOCs emissions were achieved in Jiangsu during
2017-2019 compared to 2015-2017, and the greater $NO_X$ emission reduction might
have led to the shift from VOC-limited to the transitional regime across the province
(Wang et al., 2021b). The emission controls thus helped limiting the total $O_3$
production. Although the reduction in precursor emissions was not able to fully offset
the effect of adverse meteorology condition, its encouraging effectiveness
demonstrated the validity of current emission control measures, and actual $O_3$ decline
can be expected with more stringent control actions to overcome the influence of
meteorological variation.

## 4. Conclusion remarks

In this study, we developed a high-resolution emission inventory of nine air pollutants
for Jiangsu 2015-2019, by integrating the improvements in methodology for different
sectors and incorporating the best available facility-level information and real-world
emission measurements. We evaluated this provincial-level emission inventory
through comparison with other studies at different spatial scales and air quality
modeling. We further linked the emission changes to various emission control
measures, and evaluated the effectiveness of pollution control efforts on the emission
reduction and air quality improvement.
Our study indicated that the emission controls indeed played an important role on
prevention and alleviation of air pollution. Through a series of remarkable actions in
NAPAPCP and TYAPFAP, the annual emissions in Jiangsu declined to varying
degrees for different species from 2015 to 2019, with the largest relative reduction at
53% for $SO_2$ and smallest at 6% for AVOCs. Regarding different periods, larger
abatement of $SO_2$ emissions was found between 2015 and 2017 but more substantial
reductions of $NO_X$, AVOCs and primary $PM_{2.5}$ between 2017 and 2019. Our estimates
in $SO_2$, AVOCs and $NH_3$ emissions were mostly smaller than or close to other studies,
while those for $NO_X$ and primary $PM_{2.5}$ were less conclusive. The main reasons for





the discrepancies between studies included the modified methodologies used in this
work (e.g., the procedure-based approach for AVOCs and the agricultural process
characterization for NH$_3$) and the varied depths of details on emission source
investigation (e.g., the penetrations and removal efficiencies of APCD). Air quality
modeling confirmed the benefit of refined emission data on predicting the ambient
levels of PM$_{2.5}$ and O$_3$, as well as capturing their interannual variations.
For 2015-2017 within NAPAPCP, the ultra-low emission retrofit on power sector was
most effective on SO$_2$ and NO$_X$ emission reduction, but the expansion of emission
controls to non-electricity sectors, including coal-fired boilers and key industries
would be more important for 2017-2019. AVOCs control was still in its initial stage,
and the measures on key industrial sectors and transportation were demonstrated to be
effective. Along with the gradually reduced potential for emission reduction through
end-of-pipe treatment, adjustment of energy and industrial structures should be the
future path for Jiangsu as well as other regions with developed industrial economy.
Air quality modeling suggested worsened meteorological conditions from 2015 to
2019 in terms of PM$_{2.5}$ and O$_3$ pollution alleviation. The continuous actions on
emission reduction, however, have been taking effect on reducing PM$_{2.5}$ concentration
and restraining the growth of MDA8 O$_3$ level.
The analysis justified the big efforts and investments by the local government for air
pollution controls, and demonstrated how the investigations of detailed underlying
data could help improve the precision, integrity and continuity of emission inventories.
Such demonstration was more applicable at regional scale instead of national scale,
due to the huge cost and data gap for the latter. Furthermore, the work showed how
the refined emission data could efficiently support the high-resolution air quality
modeling, and highlighted the varying and complex responses of air quality to
different emission control efforts. Therefore, the study could shed light for other
highly polluted regions in China and worldwide, with diverse stages of economic
development and air pollution controls.
Limitations remain in the current study. Attributed to insufficient data support, there
was little improvement on emission estimation for some sources compared to previous



studies, e.g., on-road transportation and residential sector. Those sources may play an
increasingly important role on emissions and air quality along with stringent controls
on power and industrial sectors, and thus need to be better stressed in the future.
Given the limited access on emission source information, the emission data for nearby
regions around Jiangsu were not refined in this work. Such limitation might lead to
some bias in analyzing the effectiveness of emission controls on air quality, as
regional transport could account for a considerable fraction of $PM_{2.5}$ and $O_3$
concentrations. Should better regional emission data get available, more analysis
needs to be conducted to separate the effectiveness of local emission controls and
efforts from nearby regions. Due to hug computational tasks through air quality
modeling, moreover, the individual emission control measures were not directly
linked to the ambient concentration, and their effectiveness on air quality
improvement cannot be obtained in details. Advanced numerical tools, e.g., the
adjoint modeling, are recommended for further in-depth analysis.

## Data availability

The gridded emission data for Jiangsu Province 2015-2019 can be downloaded at
http://www.airqualitynju.com/En/Data/List/Datadownload

## Author contributions

CGu developed the methodology, conducted the research and wrote the draft. YZhao
and LZhang developed the strategy and designed the research, and YZhao revised the
manuscript. ZXu provided the support of air quality modeling. YWang, ZWang and
HWang provided the support of emission data processing. SXia, LLi, and QZhao
provided the support of emission data.

## Competing interests

The authors declare that they have no conflict of interest.



**Acknowledgments**
This work received support from the Natural Science Foundation of China
(42177080 and 41922052), the Jiangsu Provincial Fund on $PM_{2.5}$ and $O_3$ Pollution
Mitigation (No. 2019023), and Key Research and Development Programme of
Jiangsu Province (BE2022838).

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





## Figure captions

Figure 1. Emission trends, underlying social and economic factors. Coal consumption is achieved by Chinese Energy Statistics (National Bureau of Statistics, 2016-2020). The GDP, population, and vehicle population data come from the National Bureau of Statistics, (2016-2020). Data are normalized by dividing the value of each year by their corresponding value in 2015.

Figure 2. Anthropogenic emissions by sector and year. The species include (a) $SO_2$, (b) $NO_X$, (c) CO, (d) AVOCs, (e) $NH_3$, (f) $PM_{10}$, (g) $PM_{2.5}$, (h) BC, and (i) OC. Emissions are divided into five sectors: power, industry, transportation, residential, and agriculture.

Figure 3. Changes in emissions by sector and year. The species include (a) $SO_2$, (b) $NO_X$, (c) CO, (d) AVOCs, (e) $NH_3$, (f) $PM_{10}$, (g) $PM_{2.5}$, (h) BC, and (i) OC. The 2015 emissions are subtracted from the emission data for each year to represent the additional emissions compared to 2015 levels.

Figure 4. The city-level emissions and spatial distribution include (a) $SO_2$, (b) $NO_X$, (c) AVOCs, (d) $PM_{2.5}$, and (e) $NH_3$; and (f) the proportions of emission by different regions for 2015 and 2019. The blue line indicates the Yangtze River. The map data provided by Resource and Environment Data Cloud Platform are freely available for academic use (http://www.resdc.cn/data.aspx?DATAID=201), © Institute of Geographic Sciences & Natural Resources Research, Chinese Academy of Sciences.

Figure 5. Difference in the spatial distribution of major pollutant emissions between 2015 and 2019 for (a) $SO_2$, (b) $NO_X$, (c) $PM_{2.5}$, and (d) AVOCs. The black circles represent the locations of top 10 emitters for corresponding species in each panel. The blue line indicates the Yangtze River.

Figure 6. The ratios of BVOCs to AVOCs emissions in July: (a) 2015, (b) 2017, and (c) 2019.

Figure 7. Comparison of interannual trends with MEIC, EDGAR, and ground-based observations: (a) $SO_2$ and (b) $NO_X$ ($NO_2$).





Figure 8. Comparison of Jiangsu emissions for 2017 with MEIC and An et al. (2021).
The air pollutants from left to right are $SO_2$, $NO_X$, VOCs, $NH_3$, and $PM_{2.5}$,
respectively.
Figure 9. Contributions of individual measures to emission reductions in $SO_2$, $NO_X$,
VOCs, and $PM_{2.5}$ for 2015-2017 (the left column) and 2017-2019 (the right column).
Figure 10. The monthly averages of (a) $PM_{2.5}$ and (b) MDA8 $O_3$ from CMAQ
simulation and ground observation for January, April, July and October from 2015 to
2019. The slopes of linear regressions in the panels indicate the annual variation rates
for corresponding species.
Figure 11. The concentration changes during 2015-2017 and 2017-2019 from CMAQ
for (a) $PM_{2.5}$ and (b) $O_3$ (VEMIS and VMET: meteorological conditions and
emissions fixed at 2017 level, respectively).





# Tables

**Table 1 Annual emissions of BVOCs and AVOCs and the ratios of BVOCs to AVOCs.**

|  | Year | January | April | July | October | Annual |
|---|---|---|---|---|---|---|
| BVOCs (Gg) | 2015 | 0.0020 | 8.1 | 38.0 | 3.9 | 150.0 |
|  | 2016 | 0.0017 | 8.5 | 51.4 | 2.8 | 188.1 |
|  | 2017 | 0.0023 | 9.4 | 58.7 | 2.8 | 212.7 |
|  | 2018 | 0.0020 | 9.1 | 55.5 | 3.5 | 204.3 |
|  | 2019 | 0.0017 | 6.9 | 53.4 | 4.1 | 193.2 |
| AVOCs (Gg) | 2015 | 131.3 | 102.8 | 101.8 | 104.0 | 1348.3 |
|  | 2016 | 131.2 | 102.3 | 101.3 | 103.6 | 1346.4 |
|  | 2017 | 123.4 | 97.0 | 96.0 | 98.2 | 1342.9 |
|  | 2018 | 131.6 | 102.5 | 101.6 | 103.8 | 1306.0 |
|  | 2019 | 127.7 | 99.4 | 98.4 | 100.6 | 1271.1 |
| BVOCs/AVOCs (%) | 2015 | 0.0 | 7.9 | 37.3 | 3.8 | 11.1 |
|  | 2016 | 0.0 | 8.3 | 50.7 | 2.7 | 14.0 |
|  | 2017 | 0.0 | 9.7 | 61.2 | 2.9 | 15.8 |
|  | 2018 | 0.0 | 8.9 | 54.6 | 3.4 | 15.6 |
|  | 2019 | 0.0 | 6.9 | 54.3 | 4.1 | 15.2 |





**Table 2 Air pollutant emissions in Jiangsu and comparison with previous studies**

| | Data source | Annual air pollutant emissions (Gg·yr$^{-1}$) | | | | | | |
|---|---|---|---|---|---|---|---|---|
| | | SO$_2$ | NO$_X$ | AVOCs | NH$_3$ | CO | PM$_{10}$ | PM$_{2.5}$ |
| 2014 | Li et al. (2018) | 1002 | 1315 | 1560 | 544 | 12667 | 1761 | 779 |
| 2015 | This study | 627 | 1411 | 1348 | 468 | 7735 | 711 | 491 |
| | MEIC | 626 | 1646 | 2143 | 544 | 9059 | 595 | 444 |
| | REAS | 649 | 1343 | 2063 | 611 | 10980 | 827 | 622 |
| | Sun et al. (2018) | 1230 | 1700 | 2000 | | 13780 | | |
| | Zhang et al. (2017) | | | | 703 | | | |
| 2016 | This study | 580 | 1391 | 1346 | 452 | 7397 | 687 | 475 |
| | MEIC | 468 | 1586 | 2128 | 532 | 8191 | 516 | 388 |
| | Simayi et al. (2019) | | | 2024 | | | | |
| 2017 | This study | 416 | 1331 | 1343 | 434 | 7305 | 676 | 468 |
| | MEIC | 315 | 1538 | 2132 | 528 | 7731 | 492 | 367 |
| | An et al. (2021) | 619 | 1165 | 2056 | 1093 | 17309 | 1440 | 404 |
| 2018 | This study | 374 | 1198 | 1306 | 430 | 7252 | 670 | 462 |
| | MEIC | 336 | 1456 | 1999 | 484 | 6513 | 365 | 272 |
| | Gao et al. (2022) | 210 | 830 | 3000 | 530 | 9950 | 310 | 260 |
| 2019 | This study | 296 | 1122 | 1271 | 422 | 7163 | 565 | 411 |
| | MEIC | 311 | 1414 | 1983 | 455 | 6380 | 351 | 263 |










**Figure 1**

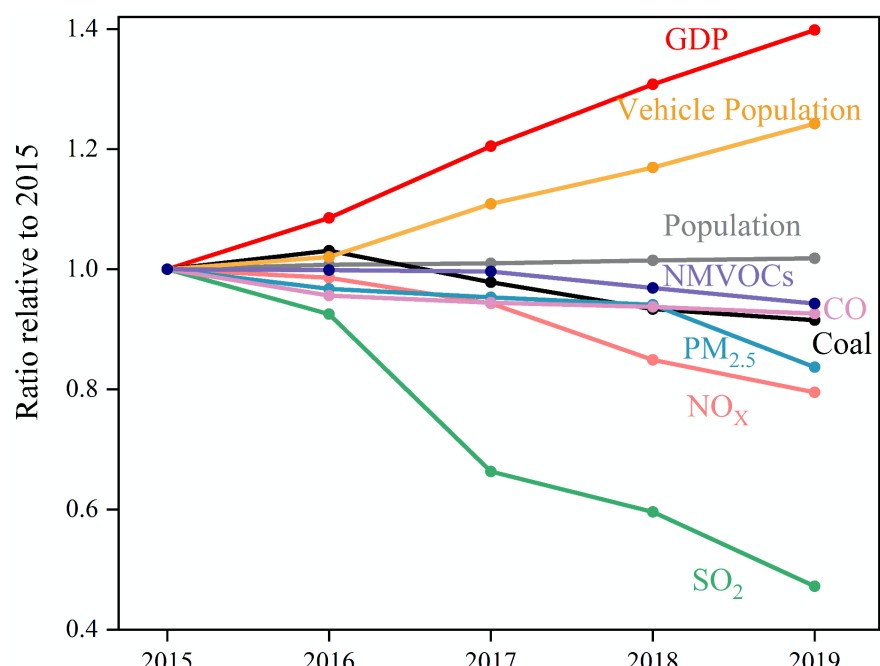



**Figure 2**

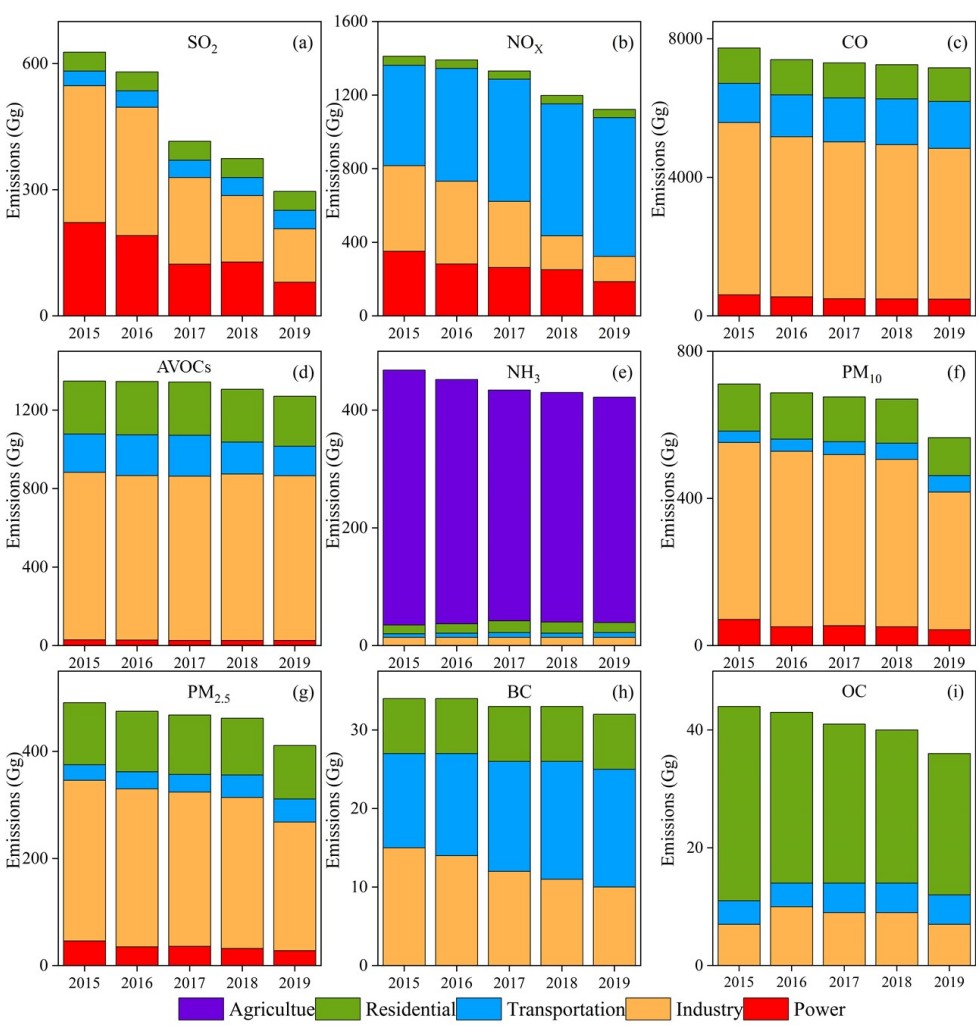











**Figure 3**

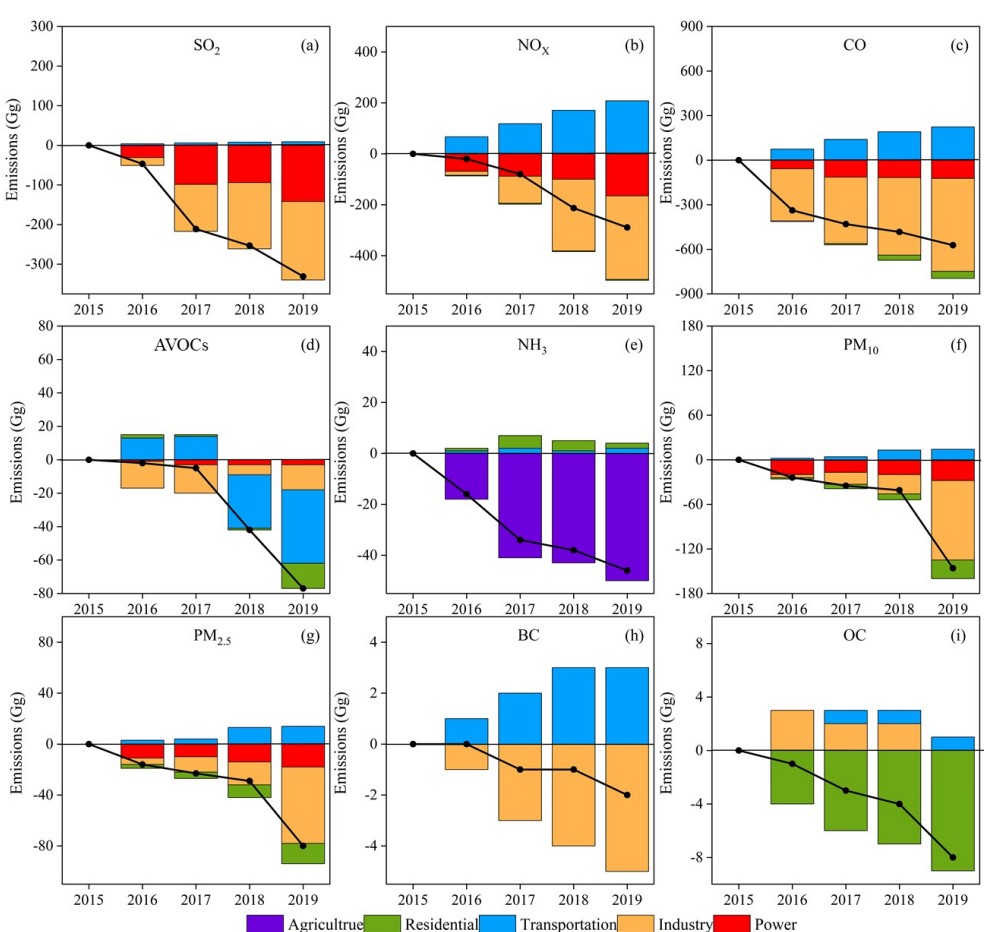










**Figure 4**

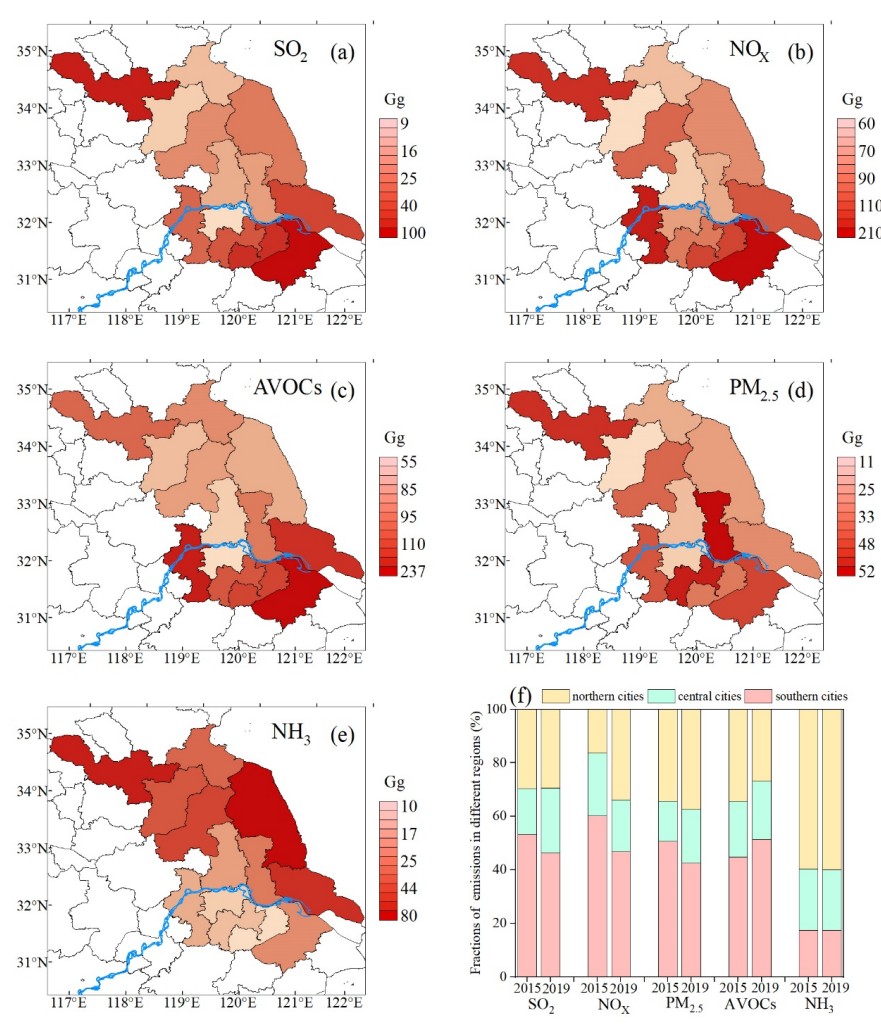




**Figure 5**

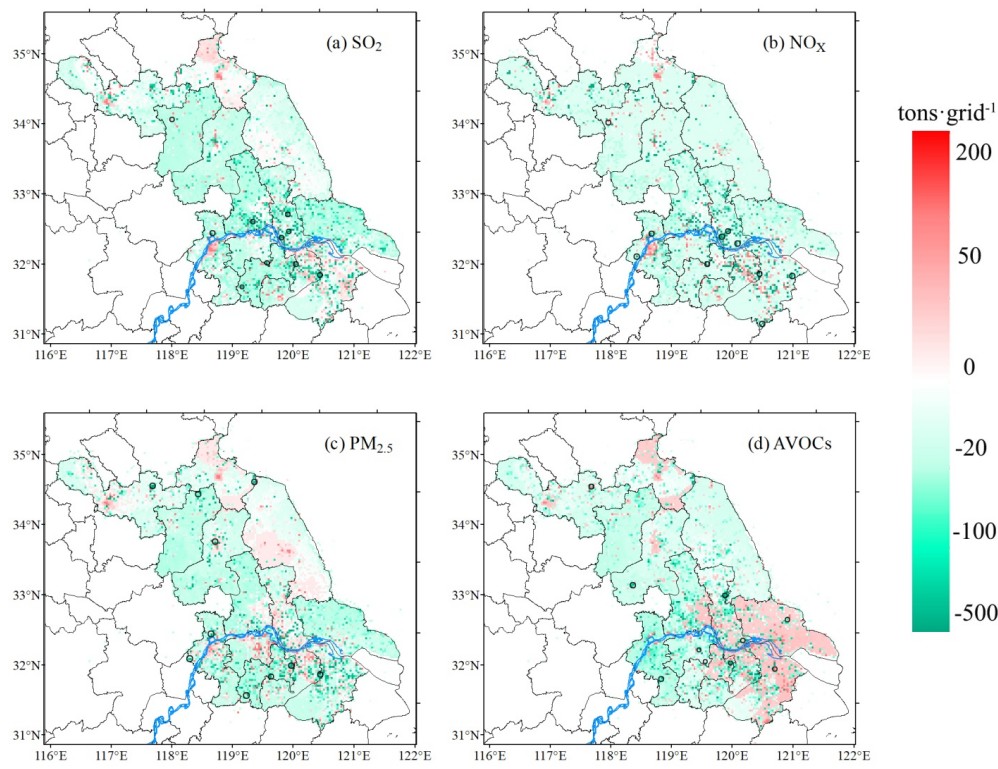







**Figure 6**

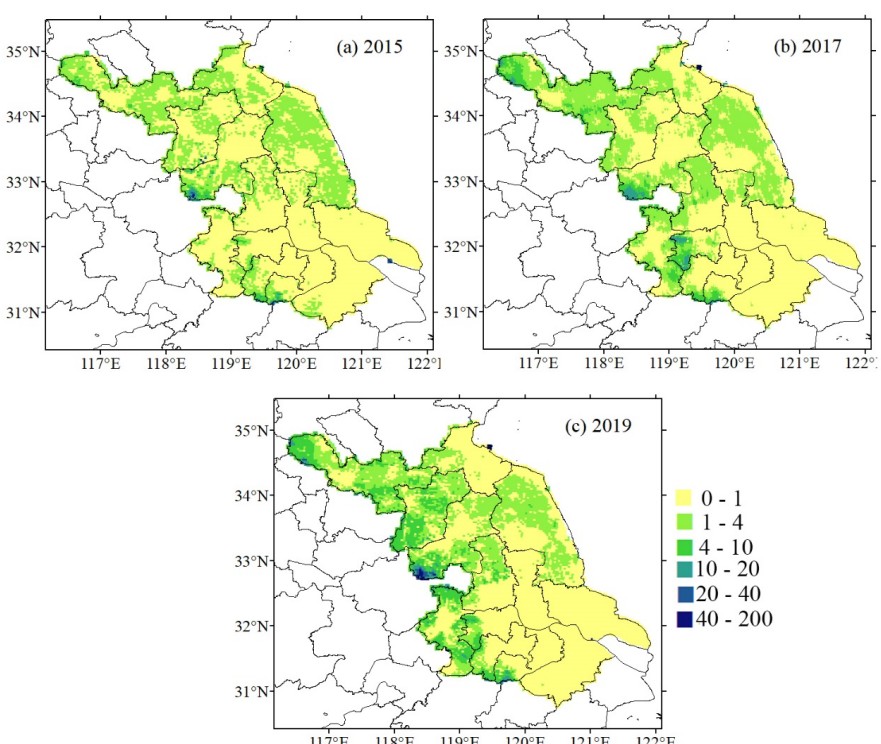











**Figure 7**

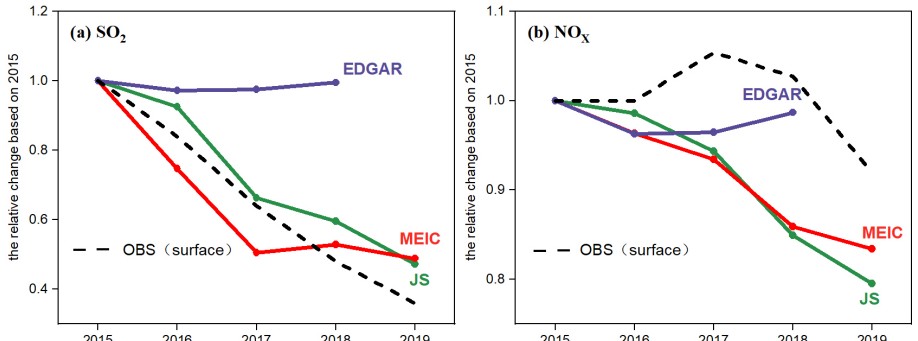




**Figure 8**

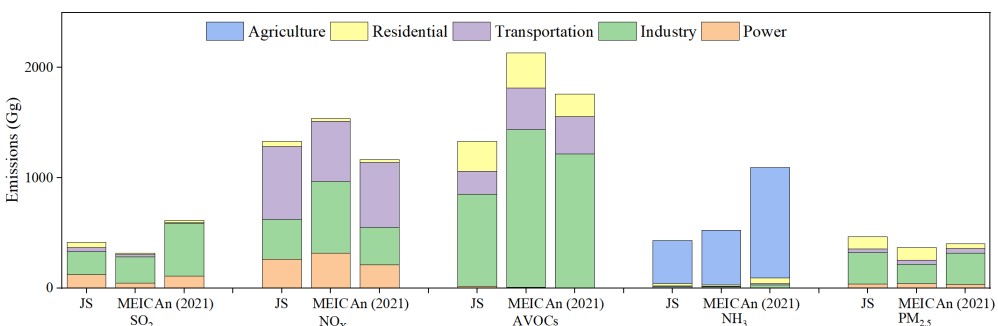












**Figure 9**





**Figure 10**

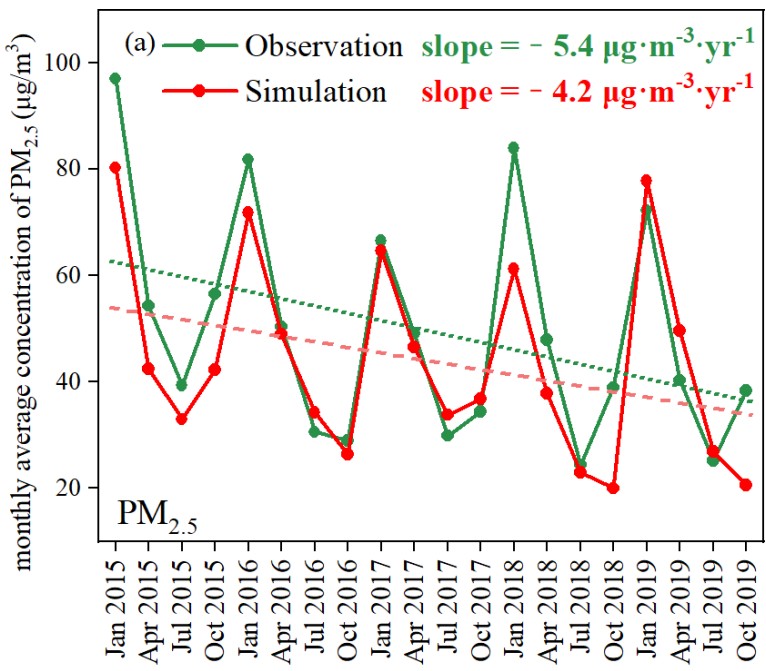

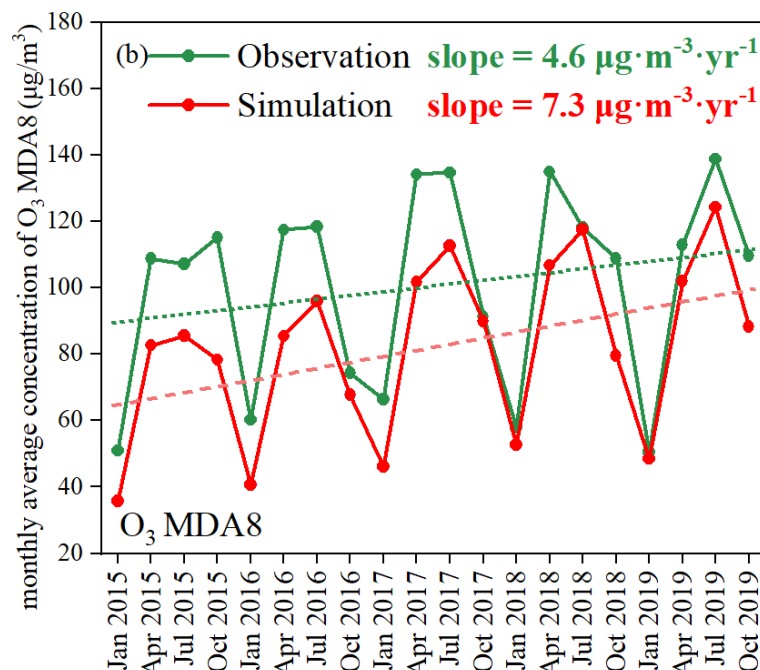








**Figure 11**

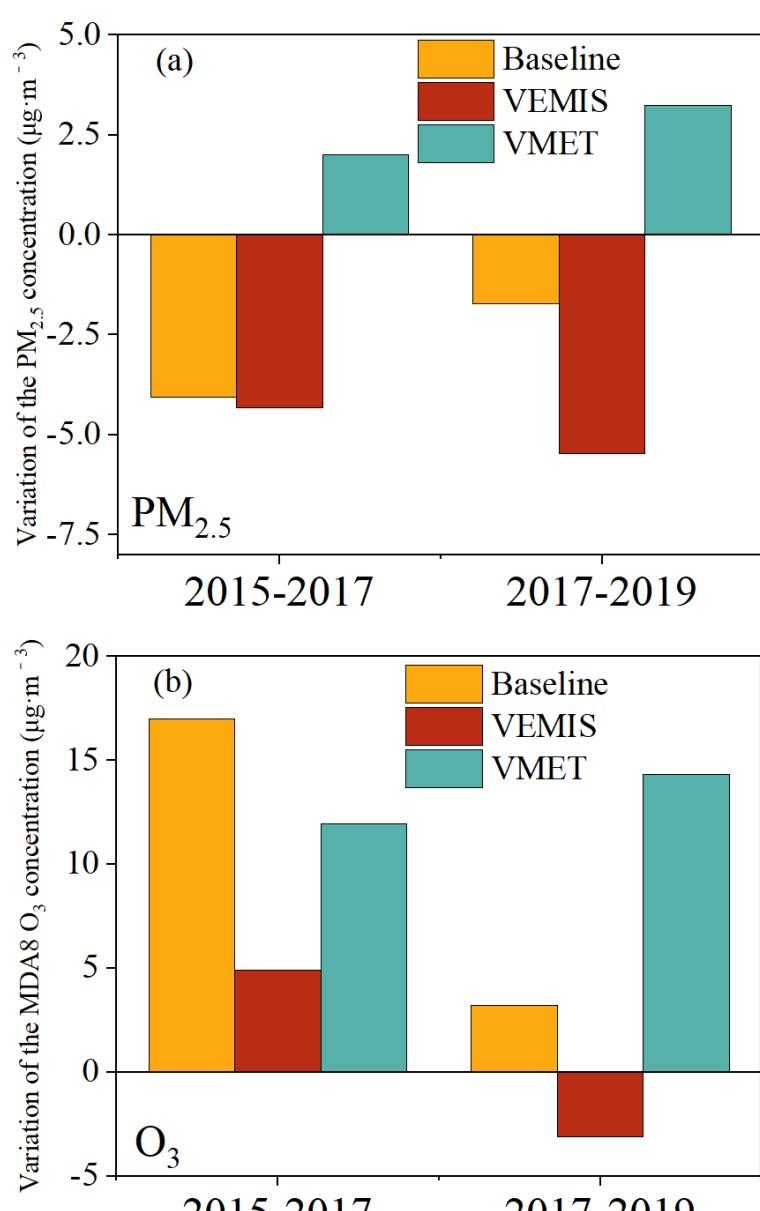
