# Peer review of "High-resolution regional emission inventory contributes to the evaluation of policy effectiveness: A case study in Jiangsu province, China"

_Atmospheric Chemistry and Physics, 2022_

## Author Response (AR1)

**Main revisions and response to reviewers' comments**

**Manuscript No.:** acp-2022-734

**Title:** High-resolution regional emission inventory contributes to the evaluation of policy effectiveness: A case study in Jiangsu province, China

**Authors:** Chen Gu, Lei Zhang, Zidie Xu, Sijia Xia, Yutong Wang, Li Li, Zeren Wang, Qiuyue Zhao, Hanying Wang, Yu Zhao

We thank very much for the valuable comments and suggestions from the reviewers, which help us improve our manuscript. The comments were carefully considered and revisions have been made in response to suggestions. Following are our point-by-point responses to the comments and corresponding revisions. **Please note that the line numbers mentioned following refer to the clean version of the revised manuscript.**

**Reviewer #1:**

*This study focuses on the development of emission inventory in the Jiangsu Province, China, during 2015-2019 based on multiple new data sources. Based on the new inventory, the influence of different policies on the changes of pollutant emissions, meteorology and emissions on the changes in PM$_{2.5}$ and O$_3$ concentrations were evaluated. Relative conclusions are helpful for further regional-level air quality improvement.*

*Overall, the manuscript is well written and its structure is well organized. Some clarifications and corrections are needed for the paper to reach the publication standard. The detailed comments are as follows ("L" indicates the line number):*

**Response and revisions:**

We thank the reviewer's valuable and positive comment, and have tried our best to make point-by-point response and revisions as summarized below.

*Q1: I highly recommend to add the comparisons of observational and CMAQ-modelling NO₂ and SO₂ concentrations. Good performance of these two species can prove the higher accuracy of the province-level emission inventory than MEIC, especially its annual changes discussed in Section 3.2.1 (L576). Besides, in the analyses of ozone underestimation, a direct evidence can be provided for NOx emission overestimation (as discussed in L790).*

**Response and revisions:**

We thank and agree the reviewer's comment very much. As we aimed at $PM_{2.5}$ and $O_3$ evaluation, we mainly compared the observational and CMAQ-modeling concentrations in the original submission, and thus did not export the results of $NO_2$ and $SO_2$ simulation based on MEIC, due to the limit of our server storage. In the revision, we collected the simulated $NO_2$ and $SO_2$ concentrations for Jiangsu based on MEIC, from the MEIC development team, Tsinghua University (the data have not been published yet), and compared them with our own simulation. A new Supplementary Table S8 has been added, which summarizes the comparison for 2017, the year with MEIC simulation data available. As can be found in the table, we can clearly find better model performance with our provincial-level inventories, and the overestimation of $NO_2$ was partly corrected indicated by the smaller NMBs with the provincial-level emission inventory. The results are exactly consistent with what the reviewer suggested.

In Section 3.4.1 of the revised manuscript, we have added the sentences: "Besides $O_3$ and $PM_{2.5}$, better model performances were also found for $SO_2$ and $NO_2$ with the provincial emission inventory than MEIC, as shown Table S8 in the Supplement. For 2017, the monthly NMB and NME ranged -38% – -24% and 43% –53% for $SO_2$, and 22% – 40% and 38% – 61% for $NO_2$. The analogous numbers for MEIC were 35% – 68% and 84% – 114% for $SO_2$, and 50% – 133% and 65% – 138% for $NO_2$, respectively (unpublished data provided by MEIC development team, Tsinghua University)." **(Lines 805-811)**. We have also added the sentence for the effect of $NO_X$

overestimation on $O_3$: "Compared to MEIC, the improved provincial emission inventory partly corrected the overestimation of $NO_X$ emissions and $NO_2$ concentrations (Table S8), and helped reduce the bias of $O_3$ concentration simulation." **(Lines 849-852).**

*Q2: If possible, it would be better to have more comparisons between all bottom-up emission inventories mentioned in this paper with official emission statistics of China/Jiangsu Province and other top-down emission inventories to make the conclusions more solid.*

**Response and revisions:**

We thank the reviewer's important comment. We have added the global emission inventory (EDGAR), official emission statistics of Jiangsu province (http://sthjt.jiangsu.gov.cn/col/col83555/index.html), and other studies including Yang et al. (2021a) and Yang et al. (2019) (a "top-down" emission estimate which was constrained with satellite observation and inverse modelling) to further evaluate the accuracy of our provincial emission inventory. The information of those emission inventories has been included in the revised Table 2. Although there were a number of studies conducted on the top-down emission inventories in China, in particular, most of them reported the values for the whole country or at relatively large spatial scales, and the results specific for Jiangsu were not available.

We have modified the sentence as: "Table 2 compares our emission estimates, by year and species, with available global (EDGAR, Crippa et al., 2020), continental (REAS, Kurokawa et al., 2020), national (MEIC), and regional emission inventories (Li et al., 2018; Sun et al.,2018; Zhang et al., 2017b; Simayi et al., 2019; An et al., 2021; Gao et al., 2022b; Yang et al., 2021a), official emission statistics of Jiangsu Province (http://sthjt.jiangsu.gov.cn/col/col83555/index.html), and an emission estimate with the "top-down" approach, i.e., constrained by satellite observation and inverse chemistry transport modelling (Yang et al., 2019)" **in lines 600-607 in the**

**revised manuscript.** For different air pollutants, we have added the comparison with official emission statistics and other newly added emission inventories in Section 3.2.1. For the $NO_X$ emission comparison, for example, we added the sentences: "The estimates of $NO_X$ emissions from MEIC, EDGAR and Sun et al. (2018) were 14-38% higher than ours, while the official emission statistics were much smaller lower than ours, attributed mainly to the absence of emissions from traffic sources in the statistics." **in lines 625-627 in the revised manuscript**, and "Constrained by satellite observation, the top-down estimation by Yang et al. (2019) was 10% and 22% smaller than our provincial emission estimation and MEIC for 2016" **in lines 639-641 in the revised manuscript**.

**References**

Yang, J., Zhao, Y., Cao, J., and Nielsen, C. P.: Co-benefits of carbon and pollution control policies on air quality and health till 2030 in China, Environ. Int., 152, 106482, https://doi.org/10.1016/j.envint.2021.106482, 2021a.

Yang, Y., Zhao, Y., Zhang, L., and Lu, Y.: Evaluating the methods and influencing factors of satellite-derived estimates of $NO_X$ emissions at regional scale: A case study for Yangtze River Delta, China, Atmos. Environ., 219, 117051, https://doi.org/10.1016/j.atmosenv.2019.117051, 2019.

*Q3: In the CMAQ modelling, the temporal and vertical profiles of emissions are also very important for a good performance. In this study, is there any improvement on these profiles, since new data applied may also provide such information?*

**Response and revisions:**

We appreciate and agree the reviewer's comment. In this work we did not make direct improvement on the temporal and vertical profiles in CMAQ (the same as MEIC), as there is still a big challenge to obtain the precise activity data of individual sources at provincial scale in China. For some species, e.g., $NH_3$, the seasonal

variation of emissions was better evaluated in this work due to application of the method characterizing agricultural processes. We have noted this limitation **in Lines in 948-952 in the revised manuscript**. Along with the gradually improving data availability, we are currently working on the development and evaluation of a "dynamic" emission inventory at the regional scale, which is expected to incorporate more details in the temporal variations of emissions and to better support air quality modeling. We will report the outcomes as soon as possible.

*Q4: Please point out the specific meanings of pollutant concentrations in the identification of meteorology and emission contributions. Are they mean pollutant concentrations in the monitoring stations? Or geographical mean values? Or population-weighted mean values? For both observational and modelling results, are the annual values both the average concentrations of four representative months of each year?*

**Response and revisions:**

We thank the reviewer's comment. In the identification of meteorology and emission contributions, the specific meaning of pollutant concentrations is the geographical mean values from CMAQ modeling, and yes, for both observation and modelling results, the annual values were average of four representative months (January, April, July, and October) of each year. We have stressed **in Line 856 in the revised manuscript**.

*Q5: For the annual changes of BVOCs emissions, are they counted as the contributions of emissions or meteorology?*

**Response and revisions:**

We thank the reviewer's important comment. In the simulations of different scenarios, the BVOCs emissions were selected in accordance with the used meteorological field for individual year. Therefore, the effect of changing

meteorology on BVOCs emissions was included in the analysis, and the interannual change in BVOCs emissions was counted as the contribution of meteorology. We have added this information **in lines 436-438 in the revised manuscript**.

*Q6: As for the writing, adverbs (e.g. significantly, largely, increasingly) are massively used in this paper. It is recommended to reduce the usage of unnecessary adverbs and be careful with the accuracy of some adverbs. For example, in L553, "biogenic sources gradually became more influential", "gradually" does not agree well the changes of BVOCs/AVOCs ratio. The tenses in some places are not correct. One example is in L571 — "it was probably due to ..." should be "it is probably due to...", since it discusses a general fact, not something happened before. Also be careful with the usage of articles — specifically, "the" should be added or deleted in many places. I pointed out some grammatical mistakes and unclear expressions in the detailed comments, but more careful checks are suggested for the authors.*

**Response and revisions:**

Many thanks for the reviewer's careful and helpful advices on the manuscript writing. Following the reviewer's comments, we have accordingly revised the manuscript and improved the language. Please find them in the following responses and the highlighted revisions in the resubmitted manuscript files.

*Detailed comments:*

*Abstract*

*Q7: L29: "in China" should be added after "province is an important administrative unit for air quality management", because this is not true for many other countries. The same is for L134 in the introduction section.*

*Q8:L38: the full names of "NMVOC", "BC" and "OC" should be used or explained.*

**Response and revisions:**

We thank the reviewer's careful comments, and the revisions have been made accordingly.

*Introduction*

*Q9:* L79: "the" should be added before "magnitude, spatial pattern, and …"

*Q10:* L95: "increasingly" should be "increasing"

*Q11:* L106: "largely weakened" is unclear. Do you mean along with the increasing diversity of emission sources, the relationships between proxies and emission distributions are weakened recently?

*Q12:* L140: "relatively" should be deleted, since there is no clear comparison.

*Q13:* L150: "it comprised" should be "it contributed to"

*Q14:* L159: "become" can be deleted.

**Response and revisions:**

Following the reviewer's suggestions, for Q9-Q10 an Q12-14, we have made the corresponding changes in the revised manuscript.

For Q11, we have added more explanation and the sentences have been modified **in the revised manuscript (Lines 107-113)** as follows:

"Such "coupling effect", however, has been demonstrated to be largely weakened for recent years. For example, a great number of big industrial facilities have been gradually moved out of urban centers, resulting in an inconsistency between emission and population hotspots. Therefore, inappropriate application of those proxies could lead to great uncertainty in emission estimation and thereby enhanced bias in air quality modeling (Zhou et al., 2017; Zheng et al., 2017)."

*Method*

*Q15: L191, L197: the number "fifty-five" and "forty-two" can be directly written as "55" and "42".*

*Q16: L192: because Table S1 also contains information on third-level emission sectors, "(see details in Table S1 in the Supplement)" can be put in the end as "(details on the first three level sectors are listed in Table S1 in the Supplement)".*

*Q17: L196: "guidelines for development of national emission inventories" => "the guidelines of national emission inventory development".*

*Q18: L205-206: "provided" => "provides"; "thus was able to considerably reduce" => "thus considerably reduces".*

*Q19: L257: "meteorological" => "meteorology"; "and" should be deleted.*

*Q20: L259-260: in "the relatively high temperature" and "the NH3 volatilization", "the" is not needed; in "NH3 volatilization for urea fertilizer use", "for" should be "from".*

**Response and revisions:**

Following the reviewer's suggestions, we have made the appropriate changes as suggested by the reviewer in the revised manuscript.

*Q21: L261: what is "metrology"?*

**Response and revisions:**

Thanks for the reminder and we are sorry for the spelling error. It should be "meteorology" and has been corrected in the revised manuscript.

*Q22: L265-274: it would be clearer to put the last sentence "in this work... for multiple years" in the beginning.*

**Response and revisions:**

Following the reviewer's suggestion, we have put the last sentence "In this work, we combined the method developed by Zhang et al. (2020) and newly tested emission factors to estimate the emissions from off-road machines in Jiangsu for multiple years." in the beginning of the paragraph **in the revised manuscript (Lines 269-271)**.

**Q23:** *L295: "we split the source profiles for some categories into finer ones" is not clear. Do you mean to use more detailed profiles for some second-level sources, instead of the more general ones for the corresponding coarser level source profiles? Also, "for example…" is suggested to be a new sentence.*

**Response and revisions:**

Thanks for reviewer's comment and yes, that's what we want to express. Following the reviewer's comment, we have clarified the sentence **in the revised manuscript (Lines 299-304)** as follows:

"Moreover, we applied more detailed profiles for some finer categories compared to the coarser source categories in the guidelines of national emission inventory development. For example, NMVOCs release in filling station into petrol and diesel release, metal surface treatment into water-based and solvent-based paints, and ink printing into offset, gravure and letterpress printing. Those efforts made the NMVOCs speciation more representative for local emission sources (Zhang et al., 2021a)."

**Q24:** *L305: "for information of stationary sources" should be "for stationary sources" or "for the information of stationary sources".*

**Q25:** *L306: "location, raw material…" => "their location, raw material".*

**Q26:** *L308, L310: "database" or "data source" should be added after "the former" and "the latter" to avoid confusion.*

**Q27:** *L328: "the" should be added before "estimation and spatial…".*

*Q28: L329-330: "with" => "by using"; for "the average emission factor by city and sector", do you mean the average emission factor of each sector in each city?*

**Response and revisions:**

We thank the reviewer's careful review and comments, and the revisions have been made accordingly.

*Q29: L331: How GDP is used to distribute the emissions?*

**Response and revisions:**

Thanks for the comment. In this work, the GDP data come from the national GDP database (1km×1km) for 2015 published by the Chinese Academy of Sciences (CAS) (https://www.resdc.cn/DOI/DOI.aspx?DOIid=33).

We have added this information **in lines 333-338 in the revised manuscript**.

*Q30: L332: "including" => "on".*

*Q31: L340-347: when introducing the control measures, it is not needed to use uppercase for the first letters. In order to avoid misunderstanding, the numbers or letters can be used like 1), 2), …*

*Q32: L349: "the" should be added before "implementation of".*

*Q33: L354-355: "it was worth noting" => "it is worth noting"; "equal" is mostly used as adjective, thus "did" should be "is".*

*Q34: L362: it should be pointed out that four months are selected to represent the four seasons.*

*Q35: L365: "the horizontal resolutions at" => "the horizontal resolutions of".*

**Response and revisions:**

We thank the reviewer's careful review and comments, and the revisions have been made accordingly.

**Q36:** *L375-376: Is the data used for the assimilation in simulations or the evaluation of modelling performance?*

**Response and revisions:**

Thanks for the reviewer's comment and we are sorry for the unclear statement. Meteorological initial and boundary conditions were obtained from the National Centers for Environmental Prediction (NCEP) datasets for the assimilation in simulations. Ground observations at 3-h intervals were downloaded from National Climatic Data Center (NCDC) to evaluate the WRF modelling performance. We have revised it **in Lines 382-388 in the revised manuscript**.

**Q37:** *L388-389: "Sp" and "Op" should be used to keep consistency with Eq. 3-4. Does "p" indicate the number of years, or the number of available data pairs?*

**Response and revisions:**

Thanks for the reviewer's comment and reminder. We have clarified the Eq.3-4 briefly **in Lines 396-399 in the revised manuscript** as follows:

$$NMB = \sum_{p=1}^{n}(S_p - O_p)/\sum_{p=1}^{n} O_p \times 100\% \qquad (3)$$

$$NME = \sum_{p=1}^{n}|S_p - O_p|/\sum_{p=1}^{n} O_p \times 100\% \qquad (4)$$

where $S_p$ and $O_p$ are the simulated and observed concentration of air pollutant, respectively, and $n$ indicates the number of available data pairs.

**Q38:** *L391-395: the discussion on modelling performance comparisons is not very clear. According to my understanding, the authors mean that the d03 modelling performance using MEIC is worse than d02 modelling performance using MEIC, thus a better d03 modelling performance using provincial emission inventory than using*

*MEIC can be expected from the comparisons in this study. Is that correct? Also, only one study is used to support the assumption. Is that universal? Can you provide more reported results of modelling performance comparison in different domains, especially for the modelling in the YRD region?*

**Response and revisions:**

We thank the reviewer's important comment. Yes, we mean that the D3 modelling performance using MEIC would be worse than D2 using MEIC (due to the extra uncertainty introduced by emission downscaling with proxy-based method), thus a better D3 modelling performance using provincial emission inventory than using MEIC can be expected from the comparisons in this work.

As mentioned by Zheng et al. (2017), the proxies of total population and GDP were poorly correlated with gridded emissions dominated by point sources, and the proxy-based methodology would result in great uncertainty in downscaling emissions and thereby air quality modeling from coarser to finer resolution. For example, a much larger bias for high-resolution simulation (additional 8-73% at 4 km) was found than that at coarser resolution (3-13% for 36 km) when MEIC was applied in predicting surface concentrations of different air pollutants. This finding suggested that provincial- and city-scale modeling efforts cannot achieve corresponding accuracies until factory-level inventories are used. Therefore, the proxy-based downscaled emissions were appropriate for modeling at the global and regional scales, while they could cause larger biases for finer resolutions. In this case, the bottom-up method must be used instead of the proxy-based method to improve the spatial representation of emission distributions.

Besides Zheng et al. (2017), other studies also suggested that the spatial errors of proxy-based emission inventories tend to increase as spatial resolutions rise (Gurney et al., 2009; Rayner et al., 2010; Oda and Maksyutov, 2011). For example, Gurney et al. (2009) highlighted the spatial biases inherent in a population-based gridded

emission inventory due to the decoupling of emissions and population at 0.1∘. The downscaling method with fixed correlation could involve large uncertainties in proxy-based gridded emissions, especially at high resolutions. Our previous work in YRD also demonstrated that downscaling national emission inventory with the proxy-based method resulted in clearly larger bias in high-resolution air quality modeling than the provincial-level emission inventory with more point source included (Zhou et al., 2017). To avoid expanding modeling bias, therefore, we did not directly downscale MEIC into the entire D3 in this work, and the improvement of provincial emission inventory could be demonstrated with better model performance (in D3) than MEIC (in D2). We have added more explanations **in Lines 400-414 in the revised manuscript**.

**References**

Gurney, K. R., Mendoza, D. L., Zhou, Y., Fischer, M. L., Miller, C. C., Geethakumar, S., and de la Rue du Can, S.: High-Resolution Fossil Fuel Combustion $CO_2$ Emission Fluxes for the United States, Environ. Sci. Technol., 43, 5535–5541, doi:10.1021/es900806c, 2009.

Oda, T. and Maksyutov, S.: A very high-resolution (1 km×1 km) global fossil fuel $CO_2$ emission inventory derived using a point source database and satellite observations of nighttime lights, Atmos. Chem. Phys., 11, 543–556, doi:10.5194/acp-11-543-2011, 2011.

Rayner, P. J., Raupach, M. R., Paget, M., Peylin, P., and Koffi, E.: A new global gridded data set of $CO_2$ emissions from fossil fuel combustion: Methodology and evaluation, J. Geophys. Res. Atmos., 115, D19306, doi:10.1029/2009JD013439, 2010.

Zheng, B., Zhang, Q., Tong, D., Chen, C., Hong, C., Li, M., Geng, G., Lei, Y., Huo, H., and He, K.: Resolution dependence of uncertainties in gridded emission

inventories: a case study in Hebei, China, Atmos. Chem. Phys., 17, 921–933, https://doi.org/10.5194/acp-17-921-2017, 2017.

Zhou, Y., Zhao, Y., Mao, P., Zhang, Q., Zhang, J., Qiu, L., and Yang, Y.: Development of a high-resolution emission inventory and its evaluation and application through air quality modeling for Jiangsu Province, China, Atmos. Chem. Phys., 17, 211–233, https://doi.org/10.5194/acp-17-211-2017, 2017.

*Q39: L399-414: it would be reader-friendly to use the tables, formulas and only some necessary explanations to state how the contributions of emission and meteorology are modeled and calculated.*

**Response and revisions:**

We thank the reviewer's important and comment. We have added a new Table S3 in the revised supplement to briefly explain how the contributions of changing emissions and meteorology were calculated, and shortened the text in the revised manuscript.

*Q40: L416: "included both from JS and nearby regions" => "is from both JS and nearby regions".*

**Response and revisions:**

We thank the reviewer's careful review and comments, and the revisions have been made accordingly.

*Results and Discussions*

*Q41: L422: "anthropogenic emissions by sector and their changes" may better summarize the contents in this section.*

**Response and revisions:**

Thanks for the reviewer's comment and we have modified the section title.

*Q42: L423: where can the information on AVOCs emissions be found? (I might be confused of NMVOCs and AVOCs emissions in this study?) The same is for the discussions in L434-436.*

**Response and revisions:**

Thanks for the reviewer's comment and we apologize for the confusing. We have revised Table S4 in the supplement (Table S3 in the original submission), and provided the information on AVOCs directly. BVOCs emissions are included in Table 2, thus not repeated in Table S4 in the revised supplement.

*Q43: L440: "grew" => "grew by".*

*Q44: L443: "clearly decoupling" is not clear.*

*Q45: L447: "accounting" => "of which the contribution accounts".*

*Q46: L471: "the" should be added before "implementation of".*

*Q47: L488-489: Table S4 can be also introduced in this sentence. I also recommend to introduce the definition of southern, northern and central cities after the sentence like "In further discussions, we classify 13 cities in Jiangsu as the southern cities (xxx), central cities (xxx) and northern cities (xxx) (their distributions are shown in Figure S1)", rather than "see the city definitions in Figure S1".*

*Q48: L493: "calculated at" => "calculated as"; "for southern, central and northern cities" => "for the southern, central and northern cities" (the same is for other places including L498).*

*Q49: L496: "were" => "are" if that is the case also for now.*

*Q50: L498: "(Figure 4)" can be deleted or written as "(Figure 4e)".*

*Q51: L506-512: the structure of this paragraph should be altered — it seems like introducing the conclusions first and then analyzing the data.*

*Q52: L513: "the" should be added before "spatial distribution of".*

**Response and revisions:**

We thank the reviewer's careful review and comments, and the revisions have been made accordingly.

*Q53: L516: "Figure 5a-c" should be changed into Figure S3 or Figure 4.*

**Response and revisions:**

Thanks for the reviewer's comment. Figure 5 provides the spatial distribution of emission change between 2015 and 2019, and the locations with super emitters. Therefore it is correct.

*Q54: L516-521: The sentence is too complex (especially, the subject of "facing" is not "more efforts"). Please express it in a more readable way.*

*Q55: L520: "opposite" is not precise according to the figures —maybe "different" is enough. "the" should be added before "spatial variation of …"*

**Response and revisions:**

Following the reviewer's comment, we have rewritten the sentence **in the revised manuscript (Lines 547-552)** as follows:

"Facing bigger challenges in air quality improvement, the economically developed southern Jiangsu has made more efforts on the emission controls of large-scale power and industrial enterprises, and achieved greater emission reduction than the less developed northern Jiangsu. Different pattern in the spatial variation of emissions was found for AVOCs (Figure 5d)."

*Q56:* *L526: "thus" is used in "there is a thus great need for substantial improvement of emission controls…", but I cannot see what is the reason for the need of emission control improvements.*

**Response and revisions:**

Thanks for the reviewer's comment. To avoid repeating word, we have deleted the second "thus" **in lines 557-559 in the revised manuscript**.

*Q57:* *L536: "season" should be deleted.*

*Q58:* *L537: "existed" => "is" or "was".*

**Response and revisions:**

We thank the reviewer's careful review and comments, and the revisions have been made accordingly.

*Q59:* *L539-541: the authors mentioned that industrial development might explain lower BVOCs emissions in the south, but what is the influence of meteorological factors? For example, higher precipitations near the Yangtze River in some seasons?*

**Response and revisions:**

Thanks for the reviewer's important comment. In this work, BVOCs emissions were estimated by MEGAN model, which is a simple mechanistic model that considers the major processes driving variations in emissions. The main meteorological factors that influence BVOCs emissions contain temperature and irradiation, and precipitation does not play a direct role in the MEGAN (Guenther et al., 2006; 2012). We extracted the simulated temperature and irradiation with WRF by season in 2017 (Figures R1 and R2), and the results show that the maximum differences in temperature and irradiation between southern and northern Jiangsu were 3-4°C and 10-25 W/m$^2$ in various seasons, respectively. According to the

sensitivity analysis for different meteorological factors and plant function type distribution by Guenther et al. (2006) and Song et al. (2012), the changes in the temperature and irradiation had a much smaller impact on the BVOCs emissions than plant function type distribution. Therefore, we did not stress it in the text.

[Figure]

Figure R1 Simulated temperature with WRF for different seasons in 2017

[Figure]

Figure R2    Simulated irradiation with WRF for different seasons in 2017

**References:**

Guenther, A., Karl, T., Harley, P., Wiedinmyer, C., Palmer, P. I., and Geron, C.: Estimates of global terrestrial isoprene emissions using MEGAN (Model of Emissions of Gases and Aerosols from Nature), Atmos. Chem. Phys., 6, 3181–3210, https://doi.org/10.5194/acp-6-3181-2006, 2006.

Guenther, A. B., Jiang, X., Heald, C. L., Sakulyanontvittaya, T., Duhl, T., Emmons, L. K., and Wang, X.: The Model of Emissions of Gases and Aerosols from Nature version 2.1 (MEGAN2.1): an extended and updated framework for modeling biogenic emissions, Geosci. Model Dev., 5, 1471–1492, https://doi.org/10.5194/gmd-5-1471-2012, 2012.

Song Y Y, Zhang Y Y, Wang Q G, An J L.: Estimation of biogenic VOCs emissions in Eastern China based on remote sensing data, Acta Scientiae Circumstantiae, 32, 2216-2227, 10.13671/j.hjkxxb.2012.09.037, 2012.

*Q60: L542: in Table 1, "%" is used for the unit of ratios, which may be misunderstood as the percentage of BVOCs in AVOCs. Thus, "×10-2" is recommended. The same is for other discussions like in L546 and L547.*

*Q61: L543: "the emission trends of both BVOCs and AVOCs" => "the trends of both BVOCs and AVOCs emissions".*

*Q62: L544: the numbers need to be consistent with what is shown in the table. Thus, "11" should be "11.1", and "16" should be "15.8".*

*Q63: L563: for this section, the main content is the comparisons between different emission inventories. "Influence of different data and methods on emission estimates" is not so direct.*

*Q64: L573: I suggest to add one sentence in the end, which is like "Therefore, we mainly compared the interannual variability of emissions in the provincial inventory and MEIC".*

*Q65: L578: "existed" => "are".*

*Q66: L581-582: "was more optimistic in..." => "describes a more optimistic...".*

**Response and revisions:**

We thank the reviewer's careful review and comments, and the revisions have been made accordingly.

*Q67: L593: I suggest to use "assessment of emission amounts" or similar items. And maybe the orders of two sections in 3.2 need to be changed — introducing emission amounts first, and then their interannual variability. The authors can consider about it.*

**Response and revisions:**

We thank and agree the reviewer's comment. Following the comment, we have rewritten the section title as "Assessment of emission amounts" and changed the order of two sections in 3.2 in the revised manuscript.

*Q68: L608: "resulted" => "results".*

*Q69: L624: when comparing numbers, "significantly" is normally used when there is statistically significant difference. Therefore, "much" might be more precise here.*

*Q70: L627: "an estimation 45% smaller" => "an estimation of 45% smaller".*

*Q71: L636: "Simaya" => "Simayi".*

**Response and revisions:**

We thank the reviewer's careful review and comments, and the revisions have been made accordingly.

*Q72: L647-648: could you please give more details on "better agreement with ground and satellite observation"?*

**Response and revisions:**

Following the reviewer's suggestion, we have added the sentence: " Compared with Infrared Atmospheric Sounding Interferometer (IASI) observation, for example, application of the emission inventory characterizing agricultural processes in CMAQ reduced the monthly NMEs of vertical column density of $NH_3$ from 44%-84% to 38%-60% in different seasons for the YRD region (Zhao et al., 2020)." **in Lines 659-663 in the revised manuscript**.

*Q73: L656: what is "thu"?*

**Response and revisions:**

Thanks for the reviewer's reminder and we are sorry for the error. The wrong word has been deleted in the revised manuscript.

*Q74: L674: the discussion in this section is a little hard to follow. Please add some hints for the readers — for example, here, it should be pointed out that the contributions of policy on $SO_2$, NOx and $PM_{2.5}$ emission changes are similar and firstly introduced.*

**Response and revisions:**

We appreciate the reviewer's important comment. We have added the hint sentences "There were some common measures for $SO_2$, $NO_X$ and $PM_{2.5}$ emission controls, thus they were discussed together below. During 2015-2017, …" **in lines 720-724 in the revised manuscript**. We have also modified the sentences as "The

driving forces of emission abatement have been changing for the three species since implementation of TYAPFAP" **in lines 754-755 in the revised manuscript,** and "Regarding AVOCs, the emission reduction…" **in lines 768-770 in the revised manuscript**.

*Q75: L722: "the" should be added before "implementation of …"*

*Q76: L739: "in a summary" => "in summary".*

*Q77: L741: "have been driving" => "have driven".*

*Q78: L758: "application of" => "applying".*

*Q79: L766: "suggested" => "suggests".*

**Response and revisions:**

We thank the reviewer's careful review and comments, and the revisions have been made accordingly.

*Q80: L772: "higher concentrations were found in summer and lower in winter" is not the case for PM$_{2.5}$.*

**Response and revisions:**

Thanks for the reviewer's careful reminder and we are sorry for the wrong description. It should be the other way: "In general, higher concentrations were found in winter and lower in summer", as we have corrected **in line 824 in the revised manuscript**.

*Q81: L777: what does "higher levels" mean? Higher concentrations or higher increase rates?*

**Response and revisions:**

We thank the reviewer's reminder, and it should be higher concentration, as we have corrected **in lines 829-830 in the revised manuscript**.

***Q82:*** *L781-783: the authors mentioned that HO₂ uptake might influence ozone changes. Besides, the effects of aerosol on radiation might be the other related to ozone changes. How these two effects are considered in CMAQ, since it is an offline model without the HO₂ uptake mechanism? If the effects of aerosol on ozone changes are not involved in the modelling, is it possible that the increase rates of annual ozone concentrations will be further overestimated when the effects of aerosol are considered?*

**Response and revisions:**

We thank and agree the reviewer's important comment. Firstly, the effects of aerosols on $O_3$ changes were not included in the CMAQ modelling. If the reduced heterogeneous absorption of HO2 by aerosols due to recent declining $PM_{2.5}$ concentration was contained in the model, it was possible that the increase rate of annual $O_3$ concentrations would be further overestimated. We have added the sentence in **lines 833-837 in the revised manuscript.** Secondly, however, as we mentioned in **lines 852-854 in the revised manuscript**, a larger underestimation in $O_3$ was revealed before 2017 compared to more recent years (Figure 8b), attributed partly to less data support on the emission sources and thereby less reliability in the emission inventory for earlier years. That is also an important reason for the higher increase rate of annual $O_3$ concentrations in our simulation compared to observation. Such overestimation is expected to be corrected for future years, if satisfying data support can be continuously guaranteed for the development of emission inventory.

***Q83:*** *L800-808: three sentences in this paragraph discuss different contents with weak connections — please revise it to make it flow. I suggest the discussion like: the annual changes of $PM_{2.5}$ and $O_3$ -> the contributions of two factors (meteorology and*

*emissions) are identified and discussed in the following -> it should be noted due to non-linearity... the mean goal is to compare the relative contributions of two factors.*

**Response and revisions:**

Following the reviewer's suggestions, we have rewritten the sentence in this paragraph **in the revised manuscript (Lines 856-865)** as follows:

"As shown in Figure 11, the provincial-level $PM_{2.5}$ concentration (geographical mean) was simulated to decrease by 4.1 μg m$^{-3}$ in 2015-2017 and 1.7 μg m$^{-3}$ in 2017-2019, and MDA8 $O_3$ increase by 17.0 μg m$^{-3}$ in 2015-2017 and 3.2 μg m$^{-3}$ in 2017-2019, in the baseline that contained the interannual changes of both anthropogenic emissions and meteorology. Smaller variations were found for more recent years for both species. With VEMIS and VMET, the contributions of the two factors were identified and discussed in the following. It should be noted that the air quality changes in baseline did not equal to the aggregated contributions in VEMIS and VMET due to non-linearity effect of the chemistry transport modeling, and the main goal of the analysis was to compare the relative contributions of the two factors."

*Q84: L817: "bigger" => "higher".*

**Response and revisions:**

We thank the reviewer's careful review and comments, and the revisions have been made accordingly.

*Supplement*

*Q85: Figure S2: be careful with the format in the first column to make is more readable.*

**Response and revisions:**

We thank the reviewer's reminder. We have redrawn Figure S2 with highlighting the policy names, adding serial numbers, and using different colors for the policies and the corresponding application sectors.

**Reviewer #2:**

*This study demonstrates the development of high-resolution emission inventory and its application on evaluating the effectiveness of emission control actions, for a typical developed and polluted province in China. The manuscript presents a sound analysis on the role of the refined emission data at the regional scale on improving the air quality simulation and supporting the pollution control evaluation. In general, the article is well organized and clearly written, with sufficient description and discussions on the relevant data and results. I have some concerns which need to be further stressed or clarified before the article can be accepted for publication.*

**Response and revisions:**

We thank the reviewer's valuable and positive comment, and have tried our best to make point-by-point response and revisions as summarized below.

*Q1: Language should be improved. Some English expression is not correct.*

**Response and revisions:**

We appreciate the reviewer's comments. We have improved the language, and tried our best to avoid grammar and spelling errors in the revised manuscript.

*Q2: I suggest adding more comparison and discussion between this study and other emission inventories (if available) to make the difference clearer.*

**Response and revisions:**

We thank the reviewer's important comment. Besides the studies included in the original submission, we have added the global emission inventory (EDGAR), official emission statistics of Jiangsu province (http://sthjt.jiangsu.gov.cn/col/col83555/index.html), and other previous studies including Yang et al. (2021a) and Yang et al. (2019) (a top-down emission inventory) to make a more comprehensive comparison with our provincial emission inventory. The information of those emission inventories has been included in the Table 2.

We have modified the sentence as: "Table 2 compares our emission estimates, by year and species, with available global (EDGAR, Crippa et al., 2020), continental (REAS, Kurokawa et al., 2020), national (MEIC), and regional emission inventories (Li et al., 2018; Sun et al.,2018; Zhang et al., 2017b; Simayi et al., 2019; An et al., 2021; Gao et al., 2022b; Yang et al., 2021a), official emission statistics of Jiangsu Province (http://sthjt.jiangsu.gov.cn/col/col83555/index.html), and an emission estimate with the "top-down" approach, i.e., constrained by satellite observation and inverse chemistry transport modeling (Yang et al., 2019)." **in lines 600-607 in the revised manuscript**. For different air pollutants, we have added the comparison with official emission statistics and other newly added emission inventories in Section 3.2.1 in the revised manuscript. For the $NO_X$ emission comparison, for example, we added the sentences: "The estimates of $NO_X$ emissions from MEIC, EDGAR and Sun et al. (2018) were 14-38% higher than ours, while the official emission statistics were much smaller lower than ours, attributed mainly to due to the absence of emissions from traffic sources in the statistics." **in lines 625-627 in the revised manuscript**, and "Constrained by satellite observation, the top-down estimation by Yang et al. (2019) was 10% and 22% smaller than our provincial emission estimation and MEIC for 2016" **in lines 639-641 in the revised manuscript**.

*Q3: Line 365-369. D3 also includes some areas of Shandong, Anhui, and Zhejiang, but the provincial inventory in this study was developed for Jiangsu. How were the emission data obtained in regions out of Jiangsu in D3? Please clarify.*

**Response and revisions:**

We appreciate reviewer's important comment. The emission data outside Jiangsu in D3 were originally from MEIC and downscaled to the resolution of 3km×3km with the "proxy-based" approach. We stressed it in **Lines 377-380 in the revised manuscript**. The approach was not perfect but can be applied for situation without detailed information on emission sources. Such application would indeed introduce some uncertainty in the simulation, which has been stressed in **Lines 952-956 in the revised manuscript**.

*Q4: Line 774. Please reword this sentence. Define "unfavorable meteorological conditions". Is the meteorology simulated by WRF inaccurate or the meteorology condition not conducive to pollution formation and accumulation?*

**Response and revisions:**

We thank and agree the reviewer's comment. We have reworded the sentence and defined the "unfavorable meteorological conditions" in **Lines 824-827 in the revised manuscript:**

"A rebound in $PM_{2.5}$ level was notably found in winter after 2017, attributed possibly to the unfavorable meteorological conditions that were more likely to exacerbate air pollution for recent years."

*Q5: Figures 2 and 3. I would like to see more analysis on the interannual variation of $PM_{2.5}$, BC and OC. For example, the reduction in OC seems bigger than BC, which could be of some significance on both air quality and climate. Could you indicate why this happened?*

**Response and revisions:**

We thank and agree the reviewer's important comment. In **Lines 504-514 in the revised manuscript**, we followed the reviewer's suggestion and added more analysis

on the interannual variations of $PM_{2.5}$, BC and OC and their climate implication as follows:

"It is worth noting that the $PM_{2.5}$ and OC emissions decreased faster than BC (Figure 2g-i). As mentioned above, the reduction in primary $PM_{2.5}$ resulted mainly from the improved energy efficiencies and emission controls in industry, and promotion of clean stoves and replacement of solid fuels with natural gas and electricity in residential sources. For OC, in particular, the largely reduced use of household biofuel and the prohibition of open biomass burning led to considerable emission abatement (18% from 2015 to 2019). However, the lack of specific APCDs and increasing heavy-duty diesel vehicles partly offset the benefit of emission controls for other sources, resulting relatively small reduction in BC emissions (6%). Besides air quality issue, the slower decline of BC than OC raised the regional climate challenge, as the former has a warming impact while the latter a cooling one."

*Q6: Figure 8. The citation format of references needs to be corrected. For example, "An (2021)" should be "An et al. (2021)."*

**Response and revisions:**

We thank the reviewer's careful review and comments, and the revisions have been made accordingly.

*Q7: Figure 9. It is indicated that Figure 9 summarizes the effect of individual measures on net emission reduction (Line 674), but the total emission reduction of all the measures seems to be greater than the net change in emissions shown in Figure S6. Please check the numbers and clarify the calculation.*

**Response and revisions:**

We thank the reviewer's comment. We have double checked the numbers and they are correct. In our study, Figure S6 shows the actual emission reduction in

2015-2019, and Figure 9 shows the net emission reduction of individual measures in 2015-2019. The aggregated emission reduction from all the measures is not equal to the actual reduction, as the factors leading to emission growth were not counted in this analysis. We have mentioned this **in Lines 361-363 in the revised manuscript.**

*Q8: It is great that the authors made detailed evaluation with CMAQ modeling and provided the results in the Supplement. However, the discussion in the main text seems descriptive. Could you be more specific on the causes of relatively big difference between observation and simulation, and also suggest the possibility for future improvement on emission estimation?*

**Response and revisions:**

We are grateful for the reviewer's comment. In this work, we found a clear underestimation in $O_3$ concentration (particularly for earlier years) through CMAQ modeling with Jiangsu emission inventory, and have mainly stressed the possible reasons (**Lines 840-849 in the revised manuscript**). In particular, we have added the comparisons of observational and CMAQ-modelling $NO_2$ concentrations (**Lines 805-811**), which provided a direct evidence for explaining the role of $NO_X$ emission overestimation on $O_3$ concentration underestimation (**Lines 849-852**). Moreover, we speculated that the more uncertain emission estimation for earlier years due to less data support could be the cause of larger overestimation in $O_3$ concentration (**Lines 852-854**).

As shown in Tables S6-S8 in the revised supplement, there was not a clear pattern for the modeling performances for individual seasons. Although the chemistry mechanisms used in the model contributed to the uncertainty, we believed it is also associated with the temporal profiles of emissions. In this work, the temporal profiles of emissions for most source categories were not improved compared to MEIC due to the difficulty in capturing the real-time variation of activity for individual emitters (e.g., the operation and energy consumption of industrial plant). We have noted the

limitation **in Lines 948-952 in the revised manuscript**. We are currently making more efforts on that issue and will report the improved emission estimates as soon as possible.

---

## Author Response (AR2)

**Main revisions and response to reviewers' comments**

**Manuscript No.:** acp-2022-734

**Title:** High-resolution regional emission inventory contributes to the evaluation of policy effectiveness: A case study in Jiangsu province, China

**Authors:** Chen Gu, Lei Zhang, Zidie Xu, Sijia Xia, Yutong Wang, Li Li, Zeren Wang, Qiuyue Zhao, Hanying Wang, Yu Zhao

We thank very much for the valuable comments and suggestions from the editor, which help us improve our manuscript. The comments were carefully considered and revisions have been made in response to suggestions. Following are our point-by-point responses to the comments and corresponding revisions. **Please note that the line numbers mentioned following refer to the clean version of the revised manuscript.**

**Editor:**

*Thank you very much for the careful revision of your manuscript. There is a number of additional improvements that need to be made before this is accepted for publication in ACP. Please perform all the following changes and resubmit a revised version with track changes.*

*Thank you for choosing ACP for the publication of your research.    The editor.*

*Note that pages and lines refer to the track changes document:*

*Page 2 line 53 between 2015 and 2017, and of NOx...*

*Page 7, line 226: 'The framework included six first-level categories this study': please correct something is missing there*

*Page 10, line 302: remove greatly*

*Page 11, line 357: replace 'finer categories' by 'second-level sources' and in the next line replace 'coarse source categories' by 'first level source categories'*

*Page 11, line 359: release during filling*

*Page 13, line 433: Four months representing the four seasons*

*Page 15, lines 505-506: 'and the anthropogenic emission variation (VEMIS)'*

*Page 17, line 583: elevate*

*Page 19, line 659: than for northern cities*

*Page 21, line 742: a broad area of Jiangsu was identified...*

*Page 23, line 889: the official statistics were much lower than ours (remove smaller)*

*Page 25, line 971: replace 'greatly' by 'significantly'*

*Page 26, line 1001: thus, they were further discussed together*

*Page 29, line 1093: as shown in Table S8*

*Page 30, line 1138: explain to what you refer as 'unfavorable meteorological conditions'. Is it temperature, precipitation, wind direction, wind speed?*

*Page 30, lines 1148: replace 'involved' by 'considered'*

*Page 30, line 1149: replace 'complicated' by 'complex'*

*Page 31, lines 1173- 1177: I suggest rephrasing as follows: ' As shown in Figure 11, in the baseline simulation that accounted for the interannual changes of both anthropogenic emissions and meteorology, the provincial-level PM2.5 concentration (geographical mean) was calculated to decrease .… in 2017-2019.*

*Page 32, line 1283: role in*

*Page 33: line 1315: such demonstrations were ...*

*Page 34, line 1337: remove 'finally'*

**Response and revisions:**

We appreciate the editor's careful review and comments, and all the suggested revisions/corrections have been made accordingly. In particular, we would like to make some more explanations on the following two comments.

*Q1: Page 11, line 357: replace 'finer categories' by 'second-level sources' and in the next line replace 'coarse source categories' by 'first level source categories'.*

**Response and revisions:**

In the guidelines of national emission inventory, there are missing source profiles in third-level or fourth-level categories, and information for coarser categories (second-level or third-level) was commonly used. Through field measurements and literature investigations, we supplement some of the missing profiles and make it possible to differentiate the finer categories. Therefore, we keep the original text in the manuscript.

*Q2: Page 30, line 1138: explain to what you refer as 'unfavorable meteorological conditions'. Is it temperature, precipitation, wind direction, wind speed?*

**Response and revisions:**

As shown in Table S2 in the supplement, both observation and simulation indicated the reduced wind speed after 2017, which could partly explain the rebound of $PM_{2.5}$ concentrations. We have added the information as "…, attributed possibly to the unfavorable meteorological conditions that were more likely to exacerbate air pollution (e.g., the reduced wind speed as shown in Table S2) for recent years" **in lines 823-825 in the revised manuscript**.